# Using Chinese Coal Gangue as an Ecological Aggregate and Its Modification: A Review

**DOI:** 10.3390/ma15134495

**Published:** 2022-06-26

**Authors:** Ying Hao, Xiaoning Guo, Xianhua Yao, Ruicong Han, Lielie Li, Min Zhang

**Affiliations:** 1School of Civil Engineering and Communication, North China University of Water Resources and Electric Power, Zhengzhou 450045, China; yinghao_ncwu@163.com (Y.H.); gxn0522@163.com (X.G.); hanruicong@ncwu.edu.cn (R.H.); lilielie@ncwu.edu.cn (L.L.); zhangmin@ncwu.edu.cn (M.Z.); 2State Key Laboratory of Eco-Hydraulics in Northwest Arid Region, Xi’an University of Technology, Xi’an 710048, China

**Keywords:** coal gangue, concrete, aggregate, solid waste utilization, modified materials, concrete performance

## Abstract

Coal gangue is a kind of industrial solid waste with serious ecological and environmental implications. Producing concrete with coal gangue aggregate is one of the green sustainable development requirements. This paper reviews the properties and preparation methods of Chinese gangue aggregate, studies the influence of gangue aggregate on concrete properties and the prediction model of gangue concrete, and summarizes the influence of modified materials on gangue concrete. The studies analyzed in this review show that different treatments influence the performance of coal gangue aggregate concrete. With the increase in the replacement ratio of coal gangue aggregate in concrete, the concrete workability and mechanical performance are reduced. Furthermore, the pore structure changes lead to decreased porosity, greatly affecting the durability. Coal gangue is not recommended for producing high-grade concretes. Nevertheless, pore structure can be improved by adding mineral admixtures, fibers, and admixtures to the coal gangue concrete. Hence, the working properties, mechanical properties, and durability of the concrete can be improved effectively, ensuring that coal gangue concrete meets engineering design requirements. Adding modified materials to coal gangue concrete is a viable future development direction.

## 1. Introduction

Coal is one of the most significant aspects of energy mining and energy production, significantly impacting the environment and endangering the sustainable utilization of coal as the primary energy source [1]. According to Yang [2], global coal production slightly decreased in 2022 compared to 2021, reaching 7.742 billion tons; however, China’s coal production increased by 1.2% over 2019. Global coal consumption and year-on-year growth rate since 2010 are shown in Figure 1. It can be seen that coal consumption in China, Europe, the United States, and India continued to grow before 2015; after 2016, coal consumption in Europe and North America decreased, whereas that in Asia continued to increase. Affected by the COVID-19 pandemic in 2020, coal consumption in other countries decreased to varying degrees except in China, Indonesia, and Vietnam. Currently, China’s coal consumption accounts for about 50% of the global total [3].

Coal gangue is a kind of hard rock produced during coal mining and, it commonly comprises limestone, claystone, and to a lesser extent, shale and volcanic rocks. It has low carbon content and generally accounts for about 10% of coal output and 25% of China’s industrial waste discharge [4,5]. China’s annual output of raw coal is 3.5 to 4 billion tons, and the annual discharge of coal gangue is about 500 to 800 million tons [6]. It is one of the largest industrial solid wastes [7]. Figure 2 shows the gangue wastes that have caused serious pollution to the water resources, land, air, and the ecological environment. Comprehensive utilization of coal gangue resources is primarily concentrated in landfills, pavement construction, power generation, brickmaking, and the chemical industries [3,8,9,10]. Although several applications have been conducted for its recovery, its utilization rate and added value are still low [11].

Concrete is a building material with the largest consumption, and its aggregate consumption is very large, accounting for about 70–80% of the concrete volume [12]. The global annual consumption of concrete is approximately 17.5 billion tons, while that of aggregate exceeds 13 billion tons [13,14]. The large-scale concrete utilization leads to the lack of natural aggregates, and the mining process has caused heavy pollution to the environment.

Therefore, considering all this, it is direly necessary to address the issues associated with aggregate sources in the construction industry [15,16]. Concrete preparation with coal gangue as aggregate can alleviate the lack of natural aggregate, which has become a new trend [17].

In this regard, how to deal with coal gangue and make it widely used in the construction industry has become essential for the development of the industry. However, there are few reviews on coal gangue concrete, especially in relation to preparation methods of coal gangue aggregate and the effect of coal gangue aggregate on concrete and the prediction model of coal gangue concrete, as well as fact that the modification materials of gangue aggregate concrete have not been summarized in detail. This paper fully analyzes and summarizes a number of studies about coal gangue aggregate concrete and achieves the following objectives: (1) to summarize the properties and preparation method of the coal gangue aggregate, (2) to study the effect of gangue aggregate on concrete performance and prediction model of coal gangue concrete, and (3) to review the studies on the influence of the modified materials on coal gangue concrete.

It is worth mentioning that most references to the use of coal gangue in the production of cementable materials have not been mentioned, since this review focuses on the use of coal gangue in aggregate. Chinese researchers mostly use coal gangue aggregate to prepare concrete and use admixtures to modify it to meet the application in different construction fields. Therefore, most relevant research was carried out by Chinese researchers. Figure 3
shows the flow diagram for the selected study.

## 2. Coal Gangue Characteristics

### 2.1. Physical Characteristics

The coal gangue shape and texture are needle flake, loose, and porous. The apparent density of Chinese coal gangue ranges from 1960 kg/m^3^ to 2760 kg/m^3^, and its maximum value is higher than the average apparent density of natural gravel of 2680 kg/m^3^. The lowest apparent density is similar to that of natural gravel, but the water absorption rate of coal gangue is higher, and the crushing value is 9.9–20.6%. The physical properties of coal gangue from different locations in China are shown as Table 1.

### 2.2. Chemical and Mineralogical Properties

According to the content of oxide in coal gangue, coal gangue can be divided into clay gangue (SiO_2_(40–70%), Al_2_O_3_(15–30%)), sandstone gangue (SiO_2_ > 70%), aluminum gangue (Al_2_O_3_ > 40%), and calcareous gangue (CaO > 30%) [25]. Table 2 shows the chemical composition and mineral composition of different types of coal gangue. Clay coal gangue is one of the most commonly used coal gangue aggregate concretes. It has high SiO_2_ and Al_2_O_3_ content, layered structure, and can produce activity under certain conditions. Figure 4 shows the X-ray diffraction (XRD) patterns of coal gangue calcined at different temperatures. It can be seen that kaolinite and quartz, the main mineral components of coal gangue, which might indicate a volcanogenic origin for these coal gangues. The mineralogical composition could also increase the applicability of coal gangue as admixture. For instance, by calcining the coal gangue, the gelling component will be activated, and the mortar strength will be improved, but the strength of the coal gangue will be reduced [26].

Table 3 shows the chemical composition of coal gangue in different areas of China. Although the SiO_2,_ Al_2_O_3_, Fe_2_O_3_, and other major components in the coal gangue from southern and northern China are not significantly different, the pozzolanic activity indices are quite different. The total alkali content controlled by geographical conditions is one of the main reasons for the pozzolanic activity difference [27]. The kaolin content of coal gangue in northern China is higher than that in southern China, which belongs to clay coal gangue (also known as coal measure kaolin). The coal gangue in the southern area of China contains more muscovite and illite minerals, and as such, it is classified as mica coal gangue [28]. In contrast, the clay coal gangue in northern China is more widely used and has a higher content of active components. The main reason for this phenomenon is that geological processes have different effects on the composition and fabric of coal gangue in different geological ages.

**Table 3 materials-15-04495-t003:** Chemical composition of coal gangue from different areas.

Sources		SiO_2_	Al_2_O_3_	Fe_2_O_3_	MgO	CaO	Na_2_O	K_2_O	TiO_2_	L.O.I	Refs
North China	Shandong	59.54	16.31	6.55	1.82	1.52	-	-	-	12.27	[34]
	Huaibei	54.12	22.38	3.56	0.71	0.98	0.64	1.13	-	-	[35]
	Liaoning	48.78	21.86	5.38	0.82	3.87	-	-	-	12.14	[36]
	Heilongjiang	58.82	27.87	8.31	-	0.78	-	1.23	1.04	1.43	[37]
	Shanxi	56.56	36.78	1.95	0.22	0.62	0.42	-	2.10	-	[38]
	Neimenggu	45.9	16.0	4.71	1.37	0.74	0.99	3.36	0.78	8.03	[39]
	Hebei	52.4	42.26	0.14	0.08	0.76	0.03	0.02	-	2.52	[40]
	Beijing	49.90	24.41	6.42	1.59	0.82	1.46	2.06	0.88	11.76	[41]
South China	Chongqing	58.10	24.50	5.31	1.05	5.73	1.14	1.54	-	-	[42]
	Guizhou	46.50	16.40	13.85	3.57	10.67	1.48	1.83		-	[27]
	Yichang	49.03	34.18	0.73	-	0.20	-	0.12	1.72	13.50	[43]
	Shanxi	52.56	16.57	3.35	2.01	1.24	0.21	2.39	-	20.71	[27]
	Jiangsu	57.95	19.02	5.32	0.82	3.16	-	-	-	-	[44]
	Jiangsu	60.24	18.50	2.58	0.52	1.48	0.14	1.53	-	-	[45]

**Figure 4 materials-15-04495-f004:**
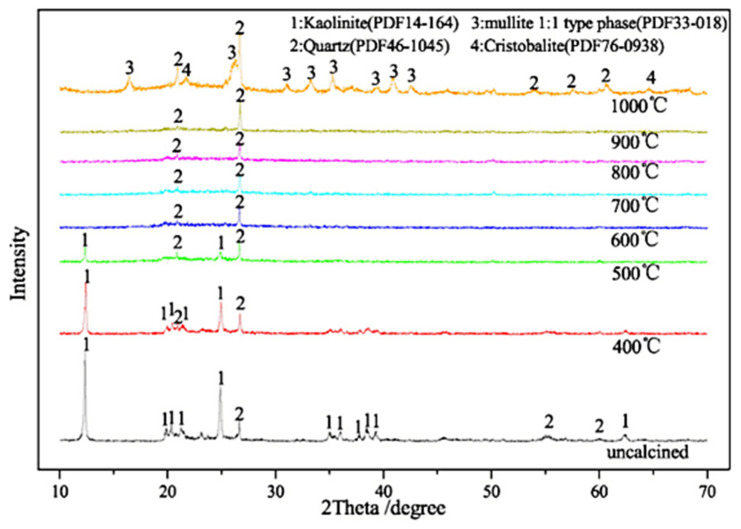
XRD patterns of coal gangue calcined at different temperatures (grinding for 30 min; holding time 2 h; heating rate 50 °C/min) [46].

## 3. Coal Gangue Aggregate Preparation

Coal gangue has a certain hardness [20]; however, it needs to be crushed to use as aggregate because of its large size. Figure 5 shows the breaking process for coal gangue with a counterattack crusher. The gangue materials are piled up around the crawler and transmitted to the counterattack crusher through the crawler. These are broken by the rotor in the middle of the confined space, and then transmitted to the large sieve shaker for screening [47]. There are three ways to use the prepared gangue. The aggregate may be used in concrete as it is, which utilizes the inherent characteristics of the aggregate or the surface of the coal gangue aggregate may be activated before mixing in concrete. The third approach is to add light aggregate ceramsite to the concrete after high-temperature calcination.

### 3.1. Direct Utilization of Coal Gangue Aggregate

#### 3.1.1. Undisturbed Coal Gangue

As shown in Figure 6, the undisturbed coal gangue is flaky and has much unwashed coal on its surface. Compared with gravel, although the undisturbed coal gangue has lower accumulation density, more pores, high water absorption, and a large crushing value, it can be used as concrete aggregate in a certain proportion. In particular, undisturbed coal gangue is suitable for producing C30 coal gangue concrete coarse and fine aggregate.

#### 3.1.2. Spontaneous Combustion Coal Gangue

The color of spontaneous combustion coal gangue will change after combustion. According to the content of iron oxide, the color will become gray white, yellow white, or red. Coal gangue is exposed to air, and much heat (self-heating) is released continuously with the oxidation exothermic reaction. It raises the temperature of the coal gangue pile until spontaneous combustion occurs [49]. The result may cause uneven heating and have a carbon removal effect, and active excitation is difficult to control. Figure 7 shows spontaneous combustion coal gangue, whose appearance is the same as undisturbed coal gangue. Nevertheless, its surface layer is mostly pottery red or pottery yellow, due to its high iron oxide content as a result of spontaneous combustion. Spontaneous combustion coal gangue has lower carbon content, higher porosity, and greater water absorption than the original coal gangue. Bulk density is between 900 kg/m^3^ and 1300 kg/m^3^ [25]. Although coal gangue activity can be improved after combustion, the pozzolanic effect in concrete is relatively small due to the uncontrollable heating temperature and limited activity [50]. The mechanical strength of concrete prepared by Spontaneous Combustion Coal Gangue was similar to that prepared by undisturbed coal gangue, and the compressive strength met the design requirements of C30 strength grade concrete [20].

### 3.2. Activation of Coal Gangue

In order to improve the low pozzolanic effect of coal gangue, appropriate activation methods are important for breaking lattice structures and increasing the amorphous phases. Mechanical activation, calcination activation, and compound activation can effectively stimulate the activity.

Mechanical activation was adopted to minimize, agglomerate, and disperse coal gangue. With this method, particle size was reduced, the specific surface area was increased, and a more uniform particle distribution was obtained, which causes lattice distortion and local destruction of minerals. Thus, the amorphous phase content was increased, and the internal energy and reactivity of coal gangue particles could be improved [51,52,53,54,55,56].

Calcination activation is conducting heat treatment in air or inert air controlled at a certain temperature. During the calcination and activation of coal gangue, the main reaction is the dehydroxylation of clay minerals such as kaolinite to form amorphous metakaolinite—an amorphous mineral with high pozzolanic activity. Equations (1)–(3) show the kaolinite calcined chemical reaction. Coal gangue has a low-temperature activation zone between 500 °C and 800 °C [39,57,58,59,60], and it has the highest activity when the calcination temperature is 700–750 °C. As the continuous temperature increased, mullite appeared at 1000 °C, and it has no hydration and does not affect gangue activation.
(1)$$Al_{2}O_{3}\cdot 2SiO_{2}\cdot 2H_{2}O\left(\mathrm{Kaolinite}\right)\to Al_{2}O_{3}\cdot 2SiO_{2}\left(\mathrm{Metakaolin}\right)+2H_{2}O$$
(2)$$2\left(Al_{2}O_{3}\cdot 2SiO_{2}\right)\left(\mathrm{Metakaolin}\right)\to 2Al_{2}O_{3}\cdot 3SiO_{2}\left(\mathrm{Alumina}-\mathrm{silicon}\;\mathrm{spinel}\right)+SiO_{2}$$
(3)$$2Al_{2}O_{3}\cdot 3SiO_{2}\left(\mathrm{Alumina}-\mathrm{silicon}\;\mathrm{spinel}\right)\to 2Al_{2}O_{3}\cdot SiO_{2}\left(\mathrm{Mullite}\right)+4SiO_{2}$$

Composite activation [56,61,62,63] uses the two kinds of activation method mentioned above to fully utilize the respective advantages. Mechanical activation of coal gangue increases the particle contact area and the amorphous content. Calcination activation can utilize a more uniform temperature to remove the carbon components in coal gangue so that more amorphous metakaolinite with pozzolanic activity can be obtained.

### 3.3. High-Temperature Calcination of Coal Gangue

High-temperature calcination at 950 °C is a kind of pottery process; ceramsite produced by high temperature calcination of coal gangue can be adopted as aggregate to prepare concrete. As shown in Figure 8, the calcined coal gangue ceramsite has rough and porous properties. Its pores are mostly derived from gases generated by high-temperature foaming components in the material [64]. By adding alternative aggregate ceramsite to the concrete, the aggregate properties and ITZ can be effectively improved, while the cohesive force can be enhanced with the rough surface. The porous nature makes its absorption rate much higher than ordinary concrete. Through the “micro pump” action, the water is slowly released during the concrete curing. The “internal curing” is carried out to reduce the unhydrated cement particles in the concrete, constantly producing hydration products and reducing the cracks caused by shrinkage.

There was a prewetting period of ceramsite aggregate during sample preparation to improve the strength of lightweight aggregate concrete and ensure that it meets the strength requirements of concrete, thus greatly reducing concrete shrinkage [65]. When the coal ceramsite volume ratio was 100%, the strength of the concrete still reached 30 MPa [66].

In summary, undisturbed coal gangue can be directly used as aggregate to prepare concrete. After combustion, the physical properties of spontaneous combustion coal gangue are reduced to a certain extent, but the direct crushing of coal gangue into aggregate still meets the requirements of preparing C30 concrete. Coal gangue has a large number of active components and high crystallinity but does not show activity. Through mechanical activation, calcination activation (700–750 °C), chemical activation, and composite activation, it can effectively improve the activity of coal gangue. High-temperature calcination (950 °C) of coal gangue can produce ceramsite, and the micro-pump effect of ceramsite can effectively improve the gangue concrete.

## 4. Performance of Coal Gangue Aggregate to Concrete

### 4.1. Working Performance

As green concrete, coal gangue aggregate concrete should have a good working performance to serve many purposes. However, coal gangue will produce more cracks on the surface after crushing. Due to the nature of porosity, the water absorption is higher than that of natural gravel aggregate, so it will absorb much water in the mixing process of concrete, which greatly influences the working performance of concrete. The slump value decreased with the increase of coal gangue replacement level, regardless of the type of concrete, i.e., ordinary concrete or high-strength concrete [18]. When the coal gangue replacement rate was 100%, the slump decreased by nearly 55% (Figure 9a). This is because the porous structure and rough surface of coal gangue particles enhance the aggregate–matrix interface and absorb more water during concrete mixing. Furthermore, the water adsorption by coal content or other organic components in coal gangue reduces the concrete workability.

In addition, calcination of coal gangue as aggregate also has an important effect on concrete fluidity. When the coal gangue aggregate is calcined at 750 °C, with the delay of calcination time, the combustion loss of coal gangue increases while the fluidity of coal gangue aggregate concrete decreases [67] (Figure 9b). The surface carbon powder and internal organic composition of coal gangue are removed by calcination, and coal gangue porosity and water absorption are increased with the increase in heat loss. Coal gangue aggregate prewetting can also improve the workability of concrete mixing, which can alleviate the decrease of fluidity caused by water absorption of coal gangue so that concrete can meet various construction requirements.

### 4.2. Mechanical Properties

#### 4.2.1. Compressive Strength

Compressive strength is the basic mechanical index of concrete performance and a judgment standard of whether the concrete can be used in engineering applications. Figure 10 shows the failure trend of concrete with coal gangue. It can be seen that cracks appear along the interface transition zone around the aggregate of the normal concrete; however, after the crushing of coal gangue concrete, cracks pass through the aggregate, and part of the aggregate breaks, which cannot resist the shear stress, leading to the reduced concrete strength [20]. Coal gangue was added with varying contents into concrete. When the content of coal gangue exceeded 70%, the concrete compressive strength, especially the early compressive strength, declined rapidly [68]. High water absorption reduced the formation of early hydration products. The water was released slowly during the curing period to promote hydration through “internal curing”. Consequently, the early compressive strength of specimens decreased, but the later strength gradually increased [69]. Yao et al. [70] prepared concrete with different mixtures of coarse coal gangue (CCG) and fine coal gangue (FCG). In the early stage, the compressive strength of coal gangue concrete was lower than that of ordinary concrete. However, with the growth of curing age, the compressive strength growth rate of coal gangue concrete was higher than that of ordinary concrete (Figure 11). Since coarse aggregate mainly bears the internal skeleton strength of concrete, replacing fine aggregate with coal gangue has a less negative impact on concrete strength. The coarse aggregate replacement may lead to the development of shear surface within the coal gangue aggregate concrete, thereby reducing concrete strength. Many studies have demonstrated that the influence of coal gangue on the strength of concrete varies with the replacement level. For example, Figure 12 shows the compressive strength of gangue concrete with different strength grades and different gangue contents. It can be seen that the factors affecting the strength of the concrete internal structure are aggregate, mortar, and the contact surface between aggregate and mortar. Aggregate is replaced by coal gangue, the aggregate strength decreases, the contact surface damage is dominant, and the strength is less affected. With the increase of strength grade, the contact surface adhesion and the strength of concrete mortar increase, and the failure is greatly affected by the aggregate. Therefore, the concrete strength decreases more after the aggregate is replaced by coal gangue.

Replacing natural gravel with coal gangue in concrete harms strength development, though coal gangue calcination can improve the strength of coal gangue aggregate concrete. The performance of coal gangue concrete with surface activation can be further improved with full activation [67]. In addition, the fuller curve R = 0.62, coal gangue aggregate density is the highest, porosity is the lowest, and concrete compressive strength is the highest [71].

#### 4.2.2. Elasticity Modulus

Figure 13 shows the stress–strain curves of coarse aggregate concrete with different coal gangue contents. It can be seen that the slope in the rising and falling stages is similar to the overall shape of the curve, while the peak stress and peak strain appear at 30–40 MPa and 0.0015–0.002, respectively. With the increase of the coal gangue replacement ratio, the slope and the peak stress decrease, indicating that the elastic modulus of concrete decreases. Figure 14 shows the elastic modulus of coal gangue concrete with different coal gangue contents, which is consistent with the conclusion of compressive strength. With the increase in replacement rate, elastic modulus decreases and negatively influences the mechanical properties. Moreover, with the improvement of concrete level, cohesive force and concrete mortar strength increase, and the damage is greatly influenced by aggregate. Therefore, the higher the concrete strength, the more the elastic modulus of concrete decreases after coal gangue replaces the aggregate.

A dimensionless model of coal gangue concrete (x = ε/ε_c_, y = σ/σ_c_) based on the stress–strain relationship test results of coal gangue concrete and the model in standard GB 50010-2010 is shown in Equation (5) [20]. Since the stress–strain relationship model of coal gangue concrete is derived from experience and is greatly affected by replacement ratio and coal gangue type, more data are needed for model validation.
(4)$$y=\left\{\begin{array}{l} ax+\left(3-2a\right)x^{2}+\left(a-2\right)x^{3},\;0\leq x\leq 1\\ \frac{x}{b{\left(x-1\right)}^{2}+x},\;x>1 \end{array}\right.$$
where a is the change of deformation modulus of concrete, and b is the area surrounded by descending section and strain axis.

The function of compressive strength was proposed to predict the elastic modulus of soil. For example, Equation (5) was recommended by ACI 318-11 (A = 4370, b = 0.50), and Equation (6) was recommended by ACI 363R (a = 3320, b = 0.50, c = 6900) [72,73]. When the replacement rate is 0%, both ACI 318-11 and ACI 363R models can predict the elastic modulus of ordinary concrete, but the prediction error will increase with the increase of replacement rate. Finally, a pair of constants was proposed to predict the elastic modulus of gangue through data verification, that is, a = 371 and b = 1.12, and the error of the constants is within ±5% [18].
(5)$$E_{c}=af_{c}^{b}$$
(6)$$E_{c}=af_{c}^{b}+c$$
where a, b, and c are constants.

**Figure 13 materials-15-04495-f013:**
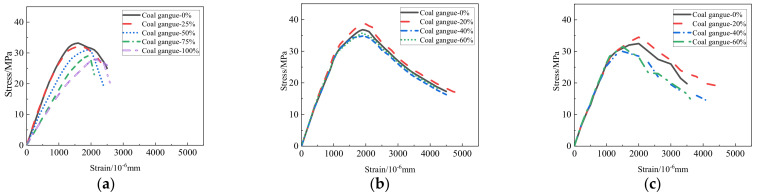
(**a**) [20], (**b**) [19], and (**c**) [74] stress-strain curves of coarse aggregate concrete with different gangue content.

**Figure 14 materials-15-04495-f014:**
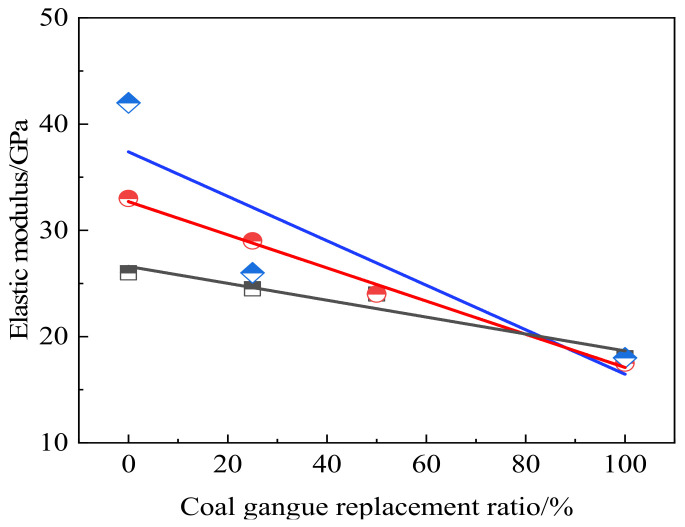
Influence of different coal gangue content on elastic modulus of coal gangue concrete [18,48].

### 4.3. Durability

#### 4.3.1. Pore Structure

Pore structure is an essential issue to study in the influence of coal gangue content on concrete durability [75]. With the increase of gangue aggregate, the porosity of a low water–binder ratio (0.39) decreases while that of a high water–binder ratio (0.50) increases. A low water–binder ratio results in a dense interfacial transition zone, which is released into the surrounding interfacial transition zone in the later stage, further promoting the hydration of cement particles. However, in the coagulated soil with a high water–binder ratio, coal gangue aggregate has a weak water regulation effect [76]. In addition, a proper increase of gangue aggregate can change the pore structure, increase the number of harmful holes, and slightly increase the pore size, reduce the compactness of concrete, and reduce the durability to a certain extent [77].

#### 4.3.2. Freezing Resistance

Water diffusion is the primary factor affecting the freeze–thaw resistance of concrete [78]. Under the freeze–thaw cycles, the water in the pores produces capillary cracks under the action of extrusion, resulting in cracks and micropores in the concrete cement matrix. As the number of freeze–thaw cycles increases, frost heave damage aggravates, the inner capillary pore wall is destroyed, and it is connected with gel pores and adjacent pores to form macrospores. When the water pressure caused by freezing and deterioration of pore structure exceeds the strength of concrete, concrete will be destroyed [79].

Compared with ordinary concrete, the compressive, tensile, and flexural strengths of coal gangue concrete decrease significantly after freeze–thaw cycles. The decline is more prominent with the increase in the coal gangue replacement ratio, which indicates that freeze–thaw causes serious damage to the interior of coal gangue concrete [80,81]. The relationship between freeze–thawing times (x) and the fracture toughness and fracture energy damage of coal gangue concrete after the freezing–thawing cycles could be fitted with the function y = ax^b^ [82]. Table 4 shows the fitting coefficient under different gangue replacement ratios. It can be seen that the higher the replacement rate of coal gangue coarse aggregate, the greater the parameter b, and the greater the freeze–thaw damage of fracture toughness and fracture energy. Moreover, the fitting power index of fracture toughness is greater than that of fracture energy, and the effect of freeze–thaw damage on fracture toughness is greater than that of fracture energy. A freeze–thaw damage evolution model of coal gangue concrete was developed and analyzed for the development mechanism of freeze–thaw damage of coal gangue concrete. It was shown that with the improvement of the replacement ratio of coal gangue concrete, the relative mass, compressive strength, and relative dynamic elastic modulus decreased rapidly [19]. Therefore, the replacement rate of coal gangue concrete should not be greater than 40% in cold areas, nor should the changes in cumulative water absorption of coal gangue concrete specimens under different replacement rates and the square root of time. The water absorption of coal gangue concrete is significantly higher than that of ordinary concrete. With the increase in the amount of coal gangue concrete, the porosity gradually increases, and the obstruction of water transmission gradually decreases. The damage of coal gangue concrete from surface to interior becomes more and more serious after freeze–thaw cycles [74].

#### 4.3.3. Carbonation Resistance

Carbonation is one of the main reasons for steel corrosion in reinforced concrete. Under normal circumstances, the reaction of CO_2_ and Ca (OH)_2_ can lead to the formation of CaCO_3_ precipitates, which reduces the alkalinity of concrete, leading to the failure of passive film on steel surface, and resulting in steel corrosion. At the same time, when the internal pH value drops to a certain extent, the hydration product C-S-H gel decomposes [83]. Coal gangue aggregate is more porous than natural aggregate. Replacing the natural aggregate concrete with coal gangue will increase porosity and accelerate the carbonation of concrete. Figure 15 shows the influence of different coal gangue contents, times, and carbide water–binder ratios on the concrete carbonation depth. The increase of coal will increase the carbonation depth of concrete and produce an adverse effect. The carbonation results are greatly influenced by the porosity of coal gangue [84]. Increasing the water–cement ratio means reducing the cement per unit volume and less Ca (OH)_2_ and C-S-H gel produced by cement hydration, resulting in many pores in cement slurry and the interface between cement slurry and aggregate. In addition, it can be seen that the carbonation depth of specimens increases with the extension of carbonation time, and the carbonation rate increases rapidly in the early stage and then slows down gradually [85].

Both compressive strength and carbonation depth of coal gangue concrete decrease with the increase of coal gangue content. Li et al. [86] proposed the relationship between carbonation depth X and strength stress σ_c_ of spontaneous combustion coal gangue concrete after curing for 28 days. The results showed a linear change, and the linear fitting empirical formula was obtained as given in Equation (7).
(7)$$X=-0.31\sigma_{c}+28.60\left(18.7\leq \sigma_{c}\leq 51.2\right)$$

The carbonation depth of coal gangue concrete is affected by many factors. A prediction (Equation (8)) based on age (t), water–cement ratio (m_w_/m_c_), coal gangue replacement ratio (r), and concrete carbonation depth (X) shows that the carbonation depth of coal gangue coarse aggregate concrete is positively correlated with the coal gangue coarse aggregate content, water–cement ratio, and the square root of carbonation time [87].
(8)$$X=0.098\sqrt{t}+50.66\frac{m_{w}}{m_{c}}+2.015r-17$$

**Figure 15 materials-15-04495-f015:**
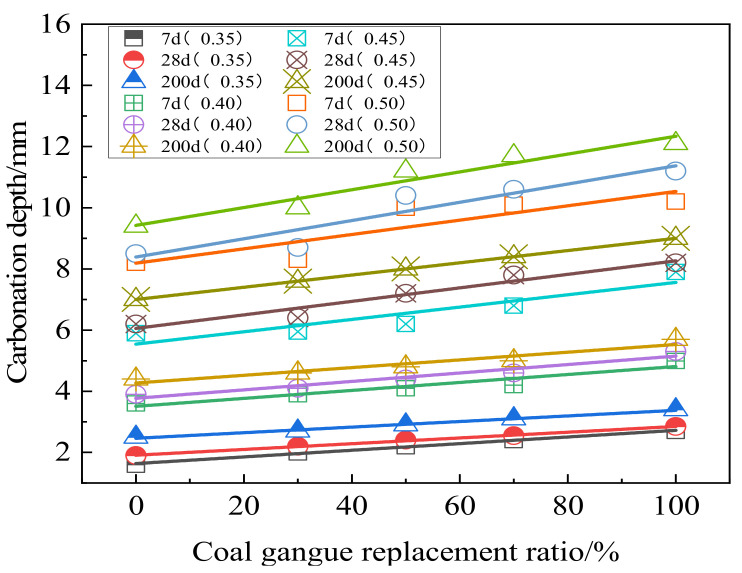
Effects of different coal gangue content, curing time, and water–binder ratio on carbonation depth of concrete [87].

#### 4.3.4. Resistance to Chloride Ion Penetration

Chloride ions intrusion may cause steel corrosion in structural concrete, resulting in concrete expansion and cracking, which seriously affects the use and safety of structural concrete. Several research studies focused on the influence of coal gangue content on the chloride ion permeability resistance of concrete [88,89]. It has been shown previously that the chloride ion permeability resistance of concrete gradually decreases with the increase of coal gangue content. The capillary pore gap is the main reason for the poor chloride ion permeability resistance of concrete. However, it will be affected by different factors in practical engineering applications, such as water–cement ratio, temperature, and crack. Figure 16a shows the influence of different penetration depths, curing temperatures, and crack widths on electric flux. Under a certain curing temperature, the chloride ion content of coal gangue concrete with three different crack widths decreases with the increase of penetration depth. When the curing temperature is high, the slope of chloride ion content is large. It shows that the higher the temperature, the more complete the cement hydration degree, the denser the gangue concrete structure, the less the electric flux increases with the depth, and the more obvious the decline in chloride ion content. Figure 16b shows the influence of different water–binder ratios and different crack widths on the chloride ion transfer coefficient. It is found that the chloride ion migration coefficient of coal gangue concrete increases with the increase of the water–cement ratio. The larger the water–cement ratio, the greater the unit volume of cement dosage reduction. Likewise, the hydration products decrease, and the capillaries at the interface between cement paste and aggregate increase, thus speeding up the transport rate of chloride ions in concrete, i.e., the permeability resistance of chloride ions increases.

#### 4.3.5. Drying Shrinkage Performance

Shrinkage is an important reason for concrete cracking, which greatly influences concrete durability. In particular, the water absorption of gangue aggregate is higher than that of natural aggregate, which negatively affects the concrete shrinkage. Figure 17a shows the concrete shrinkage performance with different water–cement ratios of coal gangue concrete. It can be seen that when the water–cement ratio is 0.45, the shrinkage of coal gangue aggregate is nearly twice that of the natural aggregate because the coal gangue aggregate absorbs more water. After being put into the drying oven, with the growth of age, the internal water evaporation of coal gangue concrete block is increased, which is the main reason for the shrinkage of gangue aggregate concrete. Concrete shrinkage can be reduced by reducing the water–binder ratio and refining the internal pores. However, an opposite trend is observed in Figure 17b: increasing the displacement rate will reduce the shrinkage of concrete, as the aggregate has a certain inherent limitation on concrete shrinkage. The coal gangue volume used in concrete mixture is larger than that of gravel, and the shrinkage strain of concrete decreases with the increase of coal gangue content.

The aggregate is prewetted before concrete preparation to reduce shrinkage of concrete. Figure 17c shows the effect of different prewetting times on coal gangue concrete shrinkage. Prewetting aggregate contains a certain amount of water; part of the water is gradually released in the later hydration process, which is an effective method to reduce concrete shrinkage [65].

In summary, coal gangue aggregate has a certain influence on concrete performance. Firstly, with the increase of coal gangue replacement, the slump gradually decreases. When coal gangue aggregate is replaced by 100%, the slump decreases by about 55%. Secondly, with the increase of gangue aggregate, the compressive strength and elastic modulus of concrete decrease. The higher the grade of concrete, the greater the negative impact. Finally, the incorporation of gangue aggregate has a negative effect on the porosity character, freezing resistance, carbonation resistance, and chloride ion invasion resistance of concrete, but it can improve the self-shrinkage of concrete by the physical characteristics of gangue aggregate itself.

## 5. Influence of Modified Materials on Coal Gangue Concrete Performance

### 5.1. Mineral Admixtures

Due to the shape effect (ball bearing effect), active effect (pozzolanic effect), and micro aggregate effect (filling effect) of the mineral admixtures, adding proper mineral admixture into gangue concrete the pore structure of cement, increase the interfacial adhesion of gangue aggregate, and improve the performance of gangue concrete.

#### 5.1.1. Fly Ash

Fly ash is generally the combusted remains of coal-fired thermal power plants. The addition of fly ash to coal gangue concrete reduces the environmental problems and helps improve the performance of coal gangue concrete. The improvement in the performance of coal gangue concrete by fly ash mainly takes advantage of its ball-bearing effect, pozzolanic effect, and filling effect, using the ball-bearing effect of fly ash to reduce the friction resistance between the particles and reduce the plastic viscosity of cement paste [93,94]. The yield was less than 25% of fly ash, the coal gangue concrete slump increased with the increase of fly ash content; with the fly ash content at more than 25%, due to the large surface area of fly ash, water plays the main role in reducing the coal gangue concrete slump. The mechanical properties of the resulting concrete depend on the strength of the hardened matrix and the properties of the interface zone of the respective aggregate [95]. The compressive, splitting tensile, flexural strength, and mechanical properties of concrete reached the maximum value when fly ash content was 30% [96]. As a fine aggregate, FA can effectively improve the gradation performance of the mixture. Short-term (3d/7d) strength increased with the increase of FA content, while long-term (28d) strength and slump increased first and then decreased [97].

Figure 18 shows the interface transition zone, interface cracks, and pores of the concrete with fly ash. When fly ash content is 0%, the interface transition zone will produce cracks extending into the aggregate due to the influence of coal gangue aggregate. When fly ash content is 30%, coal gangue concrete crack extension is interrupted by fly ash particles; with only porosity and small cracks, the bond stress of the interface transition zone is greatly improved. When the content is 40%, longer and wider cracks appear in the interface transition zone. Fly ash can effectively reduce the width of the interface transition zone of coal gangue concrete. Figure 19a shows the average micro-hardness value in the interface transition zone with different fly ash contents. The average value of micro-hardness in the interface transition zone increases first. It then decreases with the increase of fly ash content, and the average value of hardness reaches the maximum at 30% fly ash content [98]. Adding fly ash can improve the compactness of mortar to a certain extent, as confirmed by SEM images of coal gangue concrete (as shown in Figure 18), as well as solve the problem of poor durability of gangue concrete. The filling effect and pozzolanic effect of fly ash concrete make the concrete denser and more compact to reduce concrete shrinkage. Figure 19b shows the effect of different fly ash content on concrete shrinkage. It can be seen that the shrinkage of concrete can be reduced effectively with the increase of fly ash, but the mechanical properties of concrete will be affected if the content exceeds 25% [89,99]. After fly ash is added to coal gangue concrete, the chloride ion migration coefficient is negatively correlated with the content of fly ash. Filling makes coal gangue concrete denser, and secondary hydration increases the content of hydration products, and the addition of fly ash can effectively decrease the chloride ion permeability resistance of coal gangue concrete [89,99].

#### 5.1.2. Slag and Silica Fume

Silica fume and slag are common forms of waste in industry, and their output increases year by year. Their large accumulation not only occupies land resources but also aggravates environmental pollution. Therefore, it is necessary to speed up the utilization of their resources. Slag powder had adverse effects on mechanical properties and freeze resistance of coal gangue concrete [101,102]. Figure 20 shows the influence of different slag powder content and different freeze–thaw cycles on compressive strength and quality loss of coal gangue concrete. With the increase of slag powder content, the compressive strength of coal gangue concrete decreases, and the strength loss and mass loss increase after each freeze–thaw cycle. This may be due to the slow pozzolanic reaction of slag powder, less hydration product filling concrete pores, and the cement part being replaced by non-activated slag powder. The addition of silica fume can improve the spontaneous combustion coal gangue concrete. Figure 21 shows the influence of different kinds of silica fume content on the fluidity, compressive strength, and electric flux of spontaneously combusted coal gangue concrete. Adding a small amount of silica fume can improve concrete fluidity [103]. In addition, with the increasing of silica fume content, mechanical properties are improved, the electric flux is reduced, and the chloride ion resistance is enhanced, which is attributed to the combined action of silica fume filling effect and pozzolanic effect. Experiments showed that with the increase of silica fume content, the electric flux value of coal gangue concrete significantly declined, which improved the anti-chloride ion penetration ability of concrete. The addition of silica fume can improve the internal microstructure of coal gangue concrete and the composition of hydration products, improve the compactness of concrete, and increase the hindering ability of concrete on chloride ion transport [104].

### 5.2. Fiber

By increasing the fiber volume ratio, adjusting the cement matrix composition, and changing the manufacturing process, micro-crack generation and extension in the cement matrix can be controlled, and concrete durability can be improved. Compared with ordinary concrete, the strength of coal gangue aggregate concrete is insufficient, and its application is limited. Adding fiber can effectively improve the mechanical properties and durability of gangue concrete.

#### 5.2.1. Steel Fiber

Steel fiber is an important means of concrete modification. The porous structure of coal gangue leads to the permeability of coal gangue coarse aggregate concrete being higher than ordinary concrete. The distribution of theoretical pore radius and critical pore ratio is higher than that of natural coarse aggregate. The larger porosity is accompanied by a decrease in mechanical properties and durability. The addition of steel fiber can improve the interface area between coal gangue coarse aggregate and mortar matrix, change the permeability path of concrete, reduce the micro-cracks in mortar matrix, and help to improve the permeability of concrete. Figure 22 shows the influence of hook-end steel fiber [102] and corrugated steel fiber [105] with different contents on gangue concrete. Both kinds of steel fibers have an improvement effect on compressive strength. The hook-end steel fiber is easy to agglomerate, so the improvement is not as good as corrugated steel fiber, but the hook-end steel fiber is more closely bonded to concrete. The hook has the function of tension, so the tensile strength of coal gangue concrete with hook-end steel fiber is higher than that with corrugated steel fiber, but too much will also be detrimental to the tensile strength. With the increase of corrugated steel fiber content, the workability of coal gangue concrete decreases, and the compressive strength, splitting tensile strength, flexural strength, and chloride ion penetration resistance increase. When the maximum steel fiber content was 2%, the mechanical properties and chloride ion resistance were the best [102]. Shear corrugated steel fiber was added to coal gangue concrete, and the results showed that the water absorption of coal gangue concrete decreased when adding 1.0% volume fraction of steel fiber, while when 0.5% or 1.5% steel fiber was added, the water absorption increased. Three prediction models were developed: Equations (9)–(11) served as prediction models to predict the initial behavior of steel fiber-reinforced coal gangue concrete under freeze–thaw cycles. Equation (11) has the highest calculation accuracy, but the form and coefficients are complex, lacking practical significance. Equation (9) has high precision and simple form, which is suitable for engineering applications [106]. Adding a small amount of steel fiber can help improve the quality of coal gangue concrete, slow down the decline of dynamic elastic modulus, and improve the freeze resistance. Equation (12) provides a theoretical reference for the durability study of steel fiber-reinforced coal gangue concrete in a freeze–thaw environment [107].
(9)$$S_{1}=\left(0.0238+0.0136V-0.0050V^{2}\right)\times {\left(1.0364-0.0078V+0.0031V^{2}\right)}^{N}$$
(10)$$S_{1}=\left(0.0207+0.0119V-0.0045V^{2}\right)-\left(0.0016+3.8667\times 10^{-4}V-1.3333\times 10^{-4}V^{2}\right)N$$
(11)$$\begin{array}{ll} S_{1}= & \left(0.0245+0.0081V-0.0038V^{2}\right)-\left(0.0017-6.3774\times 10^{-4}V\right)N+\\ & \left(2.7315\times 10^{-4}+5.9443\times 10^{-5}V-3.2867\times 10^{-5}V^{2}\right)N^{2}-\\ & \left(5.6083\times 10^{-6}+2.3067\times 10^{-6}V-9.6667\times 10^{-7}V^{2}\right)N^{3} \end{array}$$
where S_1_ is the initial adsorption rate; N is the number of freeze-thaw cycles; V is the volume fraction of steel fiber.
(12)$$D_{\left(n\right)}=ln\left[\left(0.988-0.003f+0.101f^{2}\right)+\left(0.009-0.038f+0.092f^{2}-0.043f^{3}\right)\times n\right]$$
where n is the number of freeze–thaw cycles; D_(n)_ is the damage value of steel fiber coal gangue concrete with n freeze–thaw cycles; f is the volume fraction of steel fiber in gangue concrete.

#### 5.2.2. Other Fibers

In addition to steel fibers, polypropylene, glass, basalt, and plant fibers have different degrees of improvement on coal gangue concrete. With the increase of the volume content of polypropylene fiber, the fluidity further decreased [108]; compressive strength and bending strength reached the highest level at 6% of the content, and the drying shrinkage and cracking properties reached the lowest when the fiber content was 9%. Figure 23 shows the bending fracture diagram of coal gangue concrete mixed with polypropylene fiber. It can be seen that the appropriate addition of polypropylene fiber improves flexural performance. Mixing polypropylene fiber into concrete, the fluidity of coal gangue concrete decreased with the increase of fiber. However, the compressive, tensile, and flexural strengths of concrete were significantly improved [109].

Mechanical properties of coal gangue concrete were also effectively improved by glass fiber [110]. The mass content of glass fiber was 0.1%, the compressive strength increased slightly, and when it exceeded 1%, the compressive strength decreased. Optimum splitting tensile strength and anti-cracking performance were obtained when the mass content of glass fiber was 0.2%.

The influence of the content and length of basalt fiber on coal gangue concrete was studied and demonstrated that the compressive strength was the highest (increased by 27.5%) when the content of basalt fiber was 0.1% [111]. When the basalt fiber content was 0.15%, the splitting tensile strength was the highest—increased by 21.3%. In addition, the increase in fiber length was found to be beneficial in improving the compressive strength but not conducive to the splitting tensile strength improvement. The optimal ratio in engineering practice was determined as 0.10% fiber content and 18 mm fiber length.

Plant fiber made of straw was added to the coal gangue concrete to develop the double gangue—plant fiber recycled concrete [112]. Figure 24 shows the damaged form of the concrete test block mixed with coal gangue and plant fiber. It can be seen from Figure 24a,b that after adding fiber, the block does not peel off after crushing, while the appearance of non-fiber concrete is seriously damaged after crushing. It can be seen from Figure 24c,d that the crack width of fiber concrete is finer after fracture, and that of non-fiber concrete is wider after fracture. The addition of plant fiber can increase the concrete toughness.

The irregular distribution of fibers in the slurry can form a similar mesh structure [113,114], hindering the relative movement of particles. The higher the fiber content, the greater the impedance and the lower the mortar fluidity. Similarly, the mesh structure can help cement mortar bear tensile stress and help to improve coal gangue concrete compressive strength and splitting tensile strength. In particular, the increase of tensile strength against splitting is obvious, which leads to the increase of toughness of coal gangue concrete and the improvement of dry shrinkage and cracking. However, fiber content will lead to agglomeration, which harms the mechanical properties and enhances the extent of drying shrinkage cracking in concrete.

### 5.3. Additive

The admixtures can be added to effectively improve the performance of gangue concrete to meet the needs of different applications. A polycarboxylate ether-based superplasticizer can improve the rheological properties of coal gangue concrete. The superplasticizer was adsorbed onto the surface of cement particles to inhibit their aggregation and disintegrate them in the flocculation concrete structure and release free water [115]. It also produced the air-entraining effect, and the relative sliding between the cement particles and aggregate trend was stronger. Polycarboxylate superplasticizer was added to spontaneous combustion coal gangue concrete [116,117]. The superplasticizer could reduce the concrete porosity, improve the rheological properties of ready-mixed spontaneous combustion coal gangue aggregate concrete, increase the compressive strength, and improve the durability properties, such as frost resistance and seepage resistance.

The significance of three types of antifreeze on strength and frost resistance of coal gangue concrete is reflected in calcium chloride > calcium nitrate > ethylene glycol [118]. The effect of inorganic salt antifreeze on the water reducer was not obvious [119]. However, ethylene glycol antifreeze negatively influenced the slump of concrete mixed with water reducer. With the increase of the content of inorganic salt antifreeze, the salt freezing and thawing of concrete decreased, and a small content of ethylene glycol can enhance its salt freezing resistance. Air-entraining agents form a large number of uniform airtight bubbles inside the concrete and cut off the connected capillary channels inside the concrete, thus playing a buffer role in the expansion pressure and seepage pressure of capillary water generated by the freeze–thawing of concrete, to improve the frost resistance of coal gangue concrete. The addition of polycarboxylate superplasticizer agent can make up for the strength loss caused by more gas produced by air-entrainment agent on the concrete, and the compound use of polycarboxylate superplasticizer and air entrainment agent has a better effect [75].

The compatibility and early strength effect of three early strength agents (calcium chloride, triethanolamine, and sodium sulfate) were studied with gangue foam [120]. It was shown that none of the three early strength agents had defoaming phenomenon and had good compatibility.

In summary, the addition of modified materials can effectively improve the coal gangue aggregate on concrete, among which the morphology effect, activity effect, and micro-aggregate filling effect of mineral admixture can improve the fluidity of concrete, increase the bonding force between the interface transition zone and gangue aggregate, and improve the mechanical properties and durability. Fiber can effectively control the generation and expansion of micro-cracks in cement matrix, improve the mechanical properties and durability of concrete, but excessive agglomeration will occur. Different kinds of admixture can improve the performance of concrete according to different engineering requirements.

## 6. Discussion

To sum up, coal gangue has poor physical characteristics and a large amount of silicon and aluminum elements, but it belongs to the crystalline phase and has no activity. Replacing gravel aggregate reduces the performance of concrete to varying degrees, but meets the requirements of engineering use. Two studies [25,26] reach similar conclusions, but they lack the preparation method of coal gangue, the effect of gangue aggregate on concrete performance, the performance prediction model of concrete, and the summative description of modified materials. In this paper, the preparation method of concrete was described to further explain the influence of gangue aggregate on concrete performance; the effect of gangue aggregate on concrete was studied fully; and the performance prediction model was proposed, which is beneficial to engineering application. In addition, the influence of modified materials on gangue aggregate concrete performance was summarized and described, providing ideas for further research.

It was found that structural concrete-based coal gangue has several issues that need to be resolved before its usage in structural engineering. The following research gaps have been identified.(1)Coal gangue contains trace heavy metal elements. There are few studies on the leaching of heavy metal and organic matter from coal gangue.(2)The performance of coal gangue concrete is greatly affected by the characteristics of coal gangue. A few constitutive models were developed for coal gangue with varying characteristics, so it is not possible to predict the performance of concrete well according to the characteristics of coal gangue.(3)For coal gangue-modified materials, macro materials (particle sizes in millimeters) are mostly used, whereas there is less research of nanometer materials on coal gangue concrete.(4)Due to its physical characteristics, coal gangue is mostly used to prepare low-grade concrete, and there is a lack of research on high-grade concrete.(5)There was a lack of gangue concrete in the research on practical application in engineering and construction.

## 7. Conclusions

Coal gangue has great potential to be used as concrete aggregate, so it is of strategic significance to treat it and use it as aggregate on a large scale. This paper summarized the application of coal gangue as coarse aggregate in concrete, and the following conclusions were obtained:(1)Coal gangue performance varies greatly in different regions, and the physical characteristics of coal gangue aggregate are worse than natural gravel aggregate. As a concrete aggregate, coal gangue aggregate negatively affects the resulting mechanical properties. Its chemical components are SiO_2_, Al_2_O_3_, and Fe_2_O_3_, and its main mineral composition is kaolinite. Kaolinite has crystalline phase, and calcination can produce amorphous metakaolinite, producing activity.(2)According to the different treatment methods, undisturbed coal gangue and spontaneous combustion coal gangue are crushed directly as aggregate, which has very low activity, and the strength is lower than gravel. The aggregate activity and strength can be improved by mechanical activation and calcination activation; however, such approaches reduce concrete workability. In addition, according to the pottery process, coal gangue is burned into ceramsite. The internal curing of concrete is carried out using the “micro pump” principle to improve the internal structure of concrete, enhancing the concrete’s mechanical properties. However, it will reduce concrete fluidity.(3)With the increase of the coal gangue replacement rate, the concrete slump decreases. When the coal gangue replacement rate reaches 100%, the concrete slump can be reduced by about 55%. The increase in gangue replacement rate reduces the mechanical properties of gangue concrete, such as compressive strength, splitting tensile strength, and elastic modulus. The higher the concrete strength grade, the more negative the effect that the gangue aggregate has on the mechanical properties of concrete. In addition, an empirical formula is established to effectively predict tensile strength and splitting elastic modulus. The higher water absorption of gangue aggregate increases the extent of drying shrinkage cracking in concrete, and adversely affects the frost resistance, carbonization resistance and chloride penetration resistance of concrete. Therefore, the durability of gangue concrete can be effectively improved by decreasing the water–cement ratio, increasing curing temperature, decreasing crack width, and increasing aggregate prewetting time. The formulas for predicting frost resistance and carbonization resistance were established.(4)Modified materials can effectively improve the performance of the coal gangue concrete. Mineral admixture can improve concrete workability, compactness, ITZ between cementitious matrix and gangue aggregate, mechanical properties, and durability attributes through morphological effect, active effect, and micro-aggregate filling effect. Fibers can also effectively control the generation and expansion of micro-cracks in the cement matrix and improve the concrete mechanical properties and durability; however, an excessive amount may cause agglomeration resulting in adverse effects. Different admixtures can improve concrete performance, mechanical properties, and durability, suitable for different environmental requirements.(5)Compared with other scholars, this paper added the preparation method of concrete, supplemented the performance prediction model, and summarized and described the influence of modified materials on the performance of gangue aggregate concrete. The limitations of this study lie in the precipitation of heavy metal ions in gangue aggregate concrete, the error of constitutive model, the use of nanomaterials, the lack of high-grade concrete, and its combination with engineering practice.

## 8. Outlook

Finally, the future development of gangue aggregate concrete is proposed:(1)Corresponding studies should be made, and solutions should be developed to avoid environmental pollution and a negative effect on people’s health and safety.(2)More experimental data are needed to establish the constitutive models of different coal gangue characteristics in the future.(3)Nano-scale materials such as nano-silica, graphene oxide, and nano-calcium carbonate can be added to gangue aggregate concrete, and the morphology effect, activity effect, and micro-aggregate filling effect of nano-materials are much greater. Therefore, the influence of nanomaterials on gangue concrete could be considered.(4)In the future, coal gangue high-grade concrete can be studied by combining various modified materials.(5)Combined with actual engineering, the practical application of gangue aggregate concrete can meet the application conditions.

## Author Contributions

Y.H., X.G., and R.H.: Data curation, Writing—Original draft, Formal analysis, Writing—Review and editing. X.Y. and R.H.: Writing—Review, Funding acquisition. L.L. and M.Z.: Supervision, Writing—Review. All authors have read and agreed to the published version of the manuscript.

## Figures and Tables

**Figure 1 materials-15-04495-f001:**
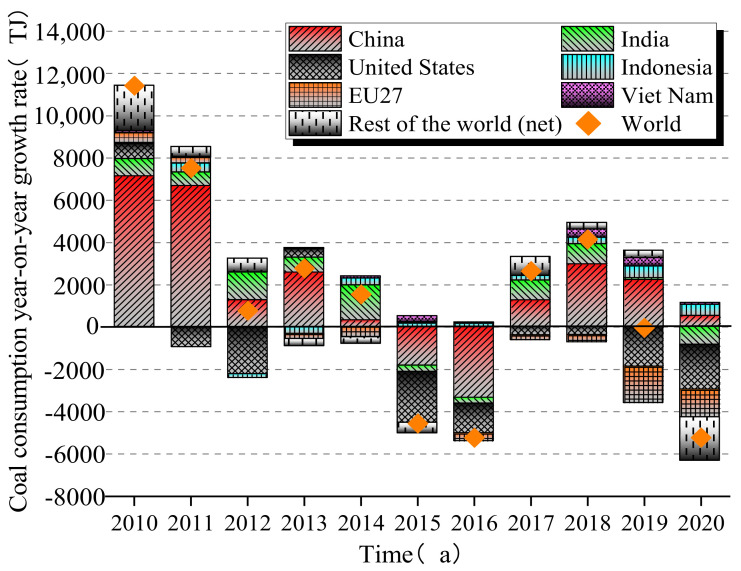
The coal consumption year-on-year growth rate of the world since 2010 (data from International Energy Agency Statistics, 2020).

**Figure 2 materials-15-04495-f002:**
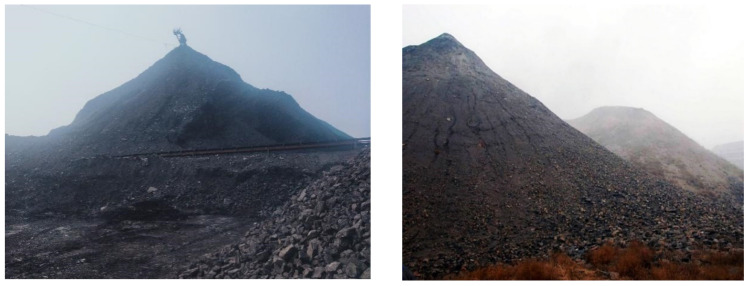
Waste coal gangue.

**Figure 3 materials-15-04495-f003:**
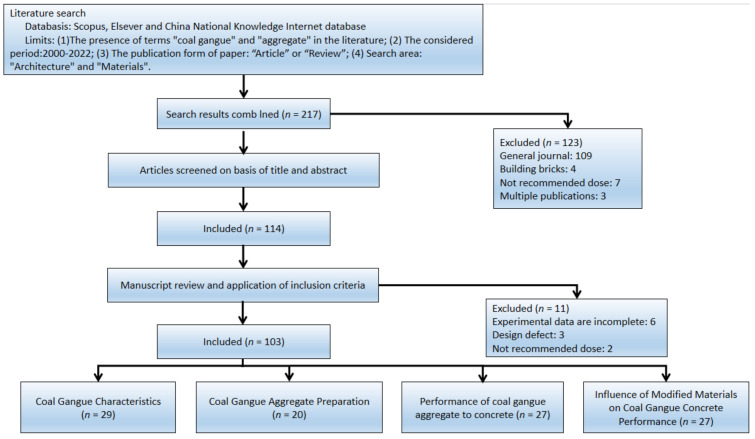
Flow diagram of the study selected.

**Figure 5 materials-15-04495-f005:**
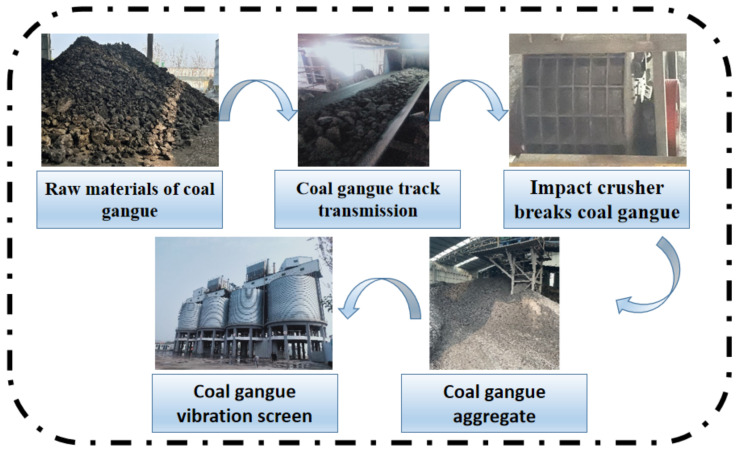
The breaking process for coal gangue with counterattack crusher.

**Figure 6 materials-15-04495-f006:**
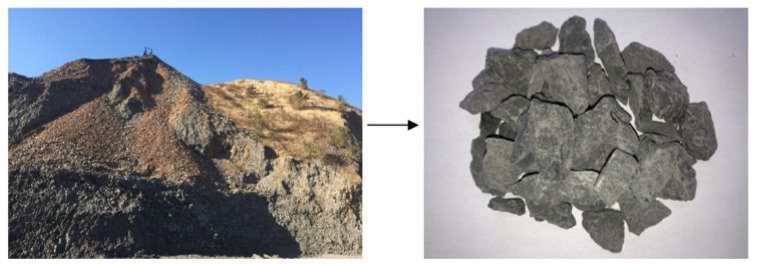
Undisturbed coal gangue [48].

**Figure 7 materials-15-04495-f007:**
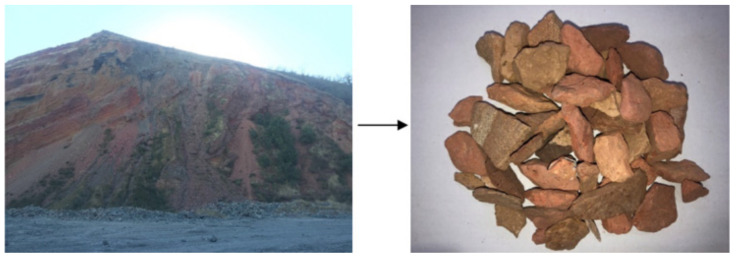
Combustion coal gangue [48].

**Figure 8 materials-15-04495-f008:**
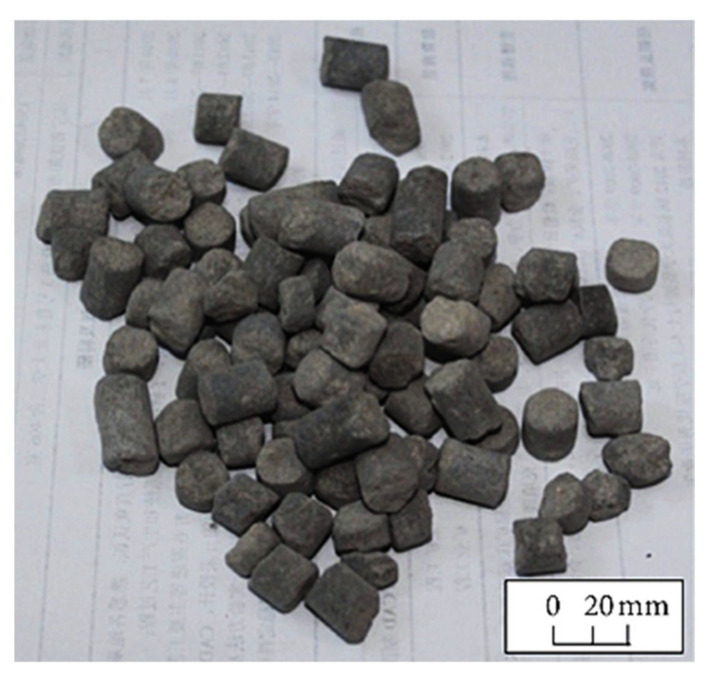
Coal gangue ceramsite [65].

**Figure 9 materials-15-04495-f009:**
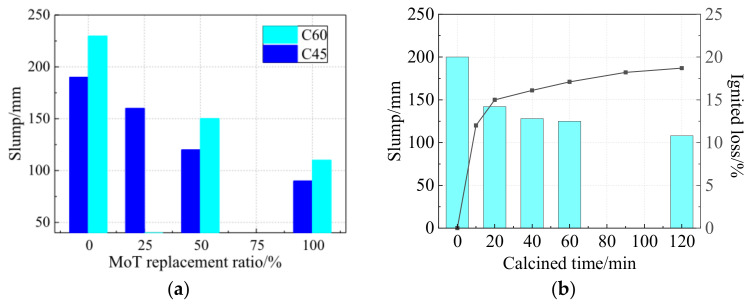
(**a**) Relationship between concrete slump and coal gangue replacement ratio [18]; (**b**) Effect of calcination time on heat loss and flow performance of coal gangue concrete [67].

**Figure 10 materials-15-04495-f010:**
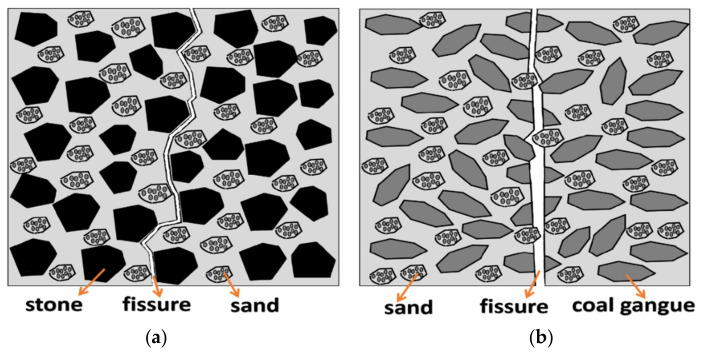
Failure trend of concrete: (**a**) Normal concrete; (**b**) Coal gangue concrete.

**Figure 11 materials-15-04495-f011:**
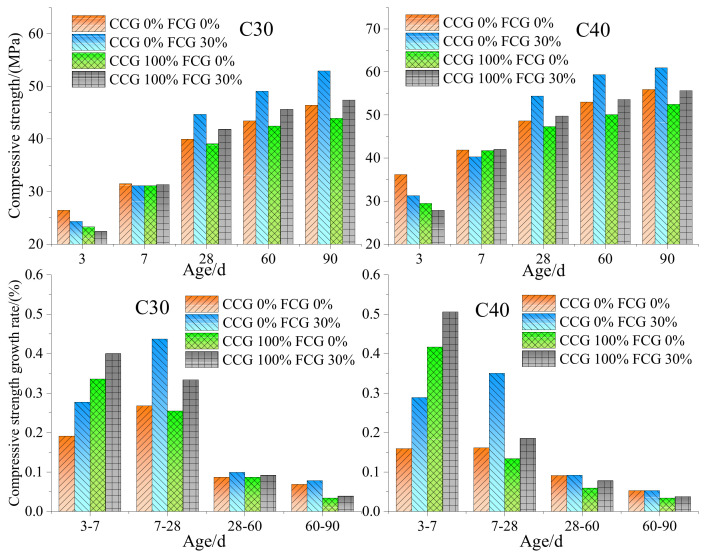
Effect of different content on compressive strength and strength growth rate of coal gangue concrete with different strength grades [70].

**Figure 12 materials-15-04495-f012:**
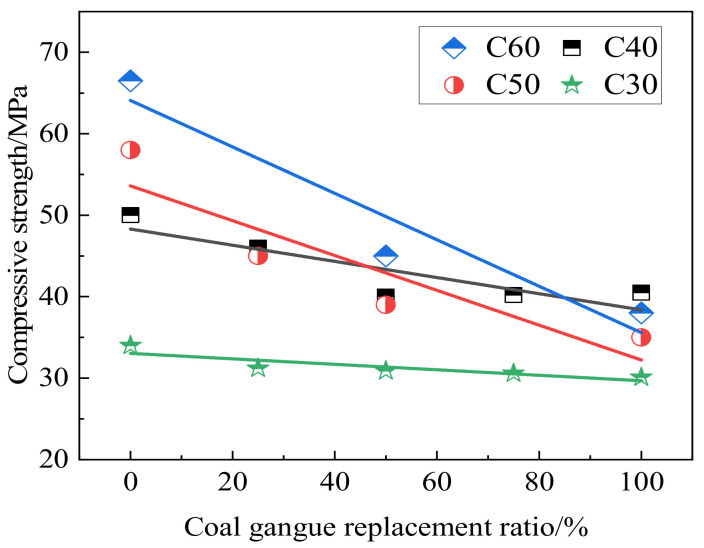
Influence of different gangue content on compressive strength of gangue concrete with different strength grades [18,20,22].

**Figure 16 materials-15-04495-f016:**
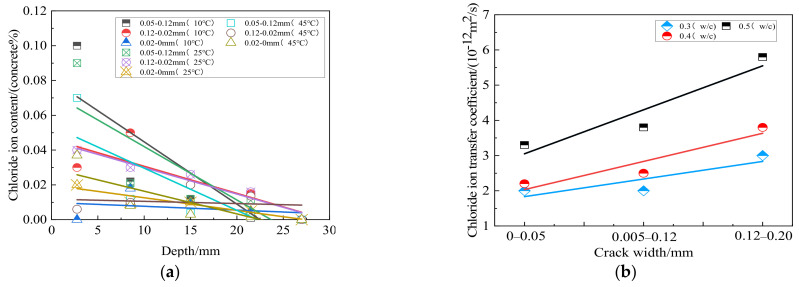
(**a**) Effects of different penetration depths, curing temperatures and crack widths on electric flux [90]; (**b**) Effects of different water–-cement ratios and crack widths on chloride ion transfer coefficient [90].

**Figure 17 materials-15-04495-f017:**
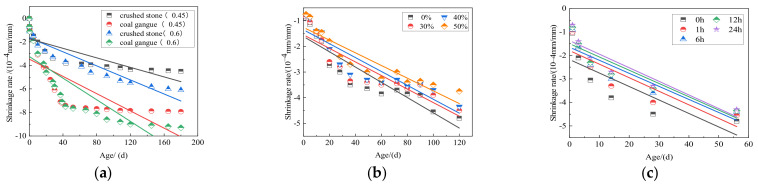
(**a**) Effect of different water-binder ratios on concrete shrinkage [91]; (**b**) Effect of different displacement rates of coal gangue on concrete shrinkage [92]; (**c**) Effect of prewetting time on shrinkage rate [65].

**Figure 18 materials-15-04495-f018:**
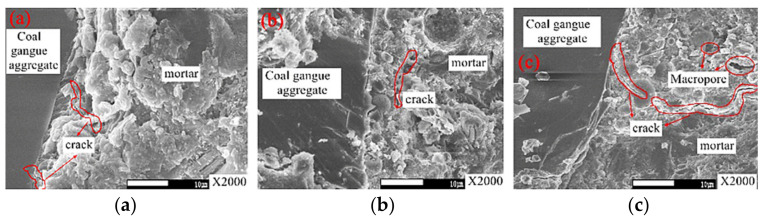
Cracks and pores in interfacial transition zone [96]: (**a**) Common coal gangue concrete, (**b**) 30% fly ash, (**c**) 40% fly ash.

**Figure 19 materials-15-04495-f019:**
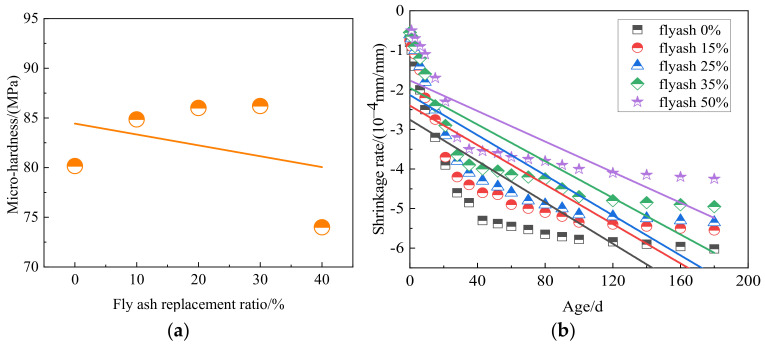
(**a**) Average value of micro-hardness in interface transition zone with different fly ash content [98]; (**b**) Effect of different fly ash content on concrete shrinkage [100].

**Figure 20 materials-15-04495-f020:**
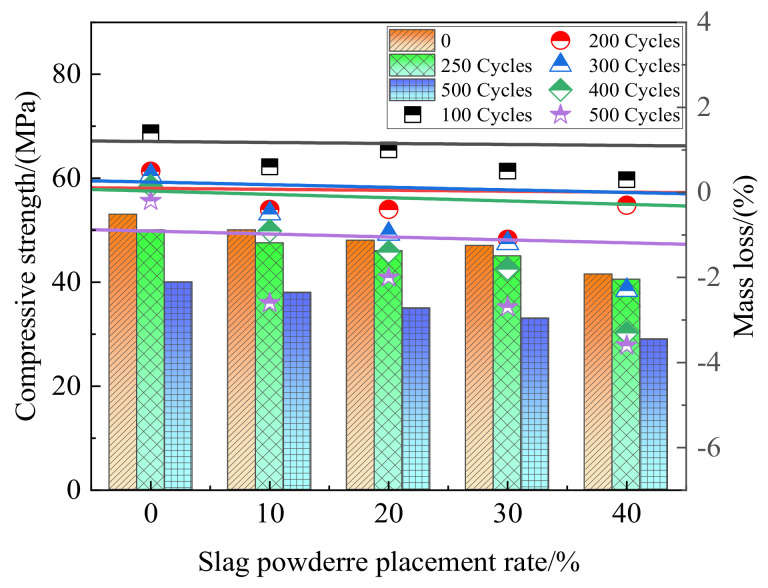
Influence of different amounts of slag powder and different freeze-thaw cycles on compressive strength and quality loss of coal gangue concrete [102].

**Figure 21 materials-15-04495-f021:**
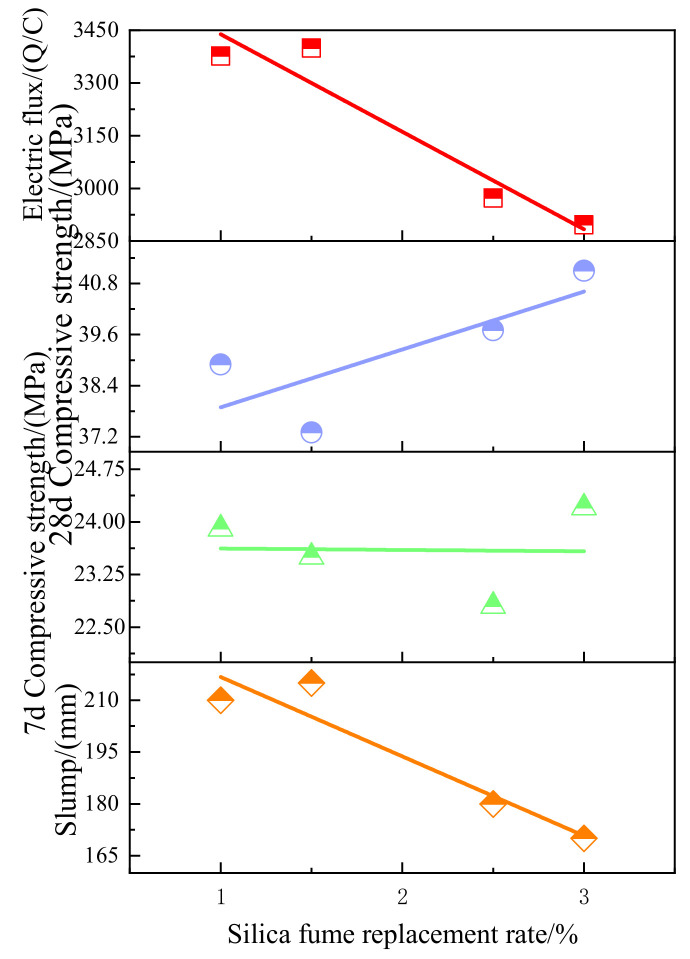
Effect of different amounts of silica fume on fluidity compressive strength and electric flux of coal gangue concrete [103].

**Figure 22 materials-15-04495-f022:**
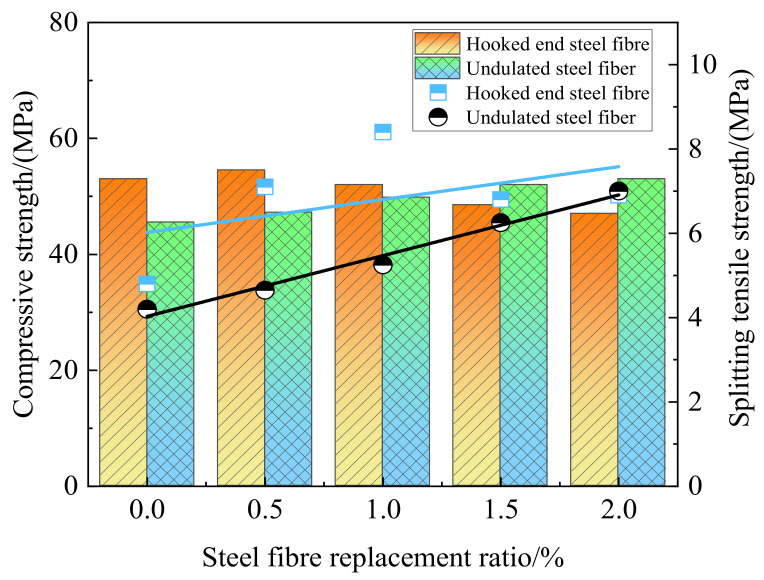
Strength of aggregate concrete mixed with gangue of different amounts of steel fiber [102,105].

**Figure 23 materials-15-04495-f023:**
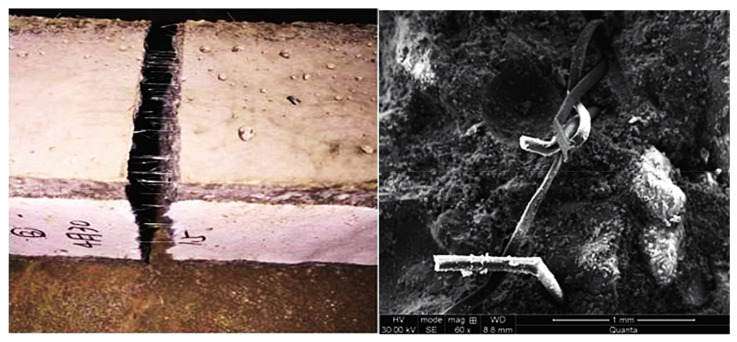
Flexural fracture of coal gangue concrete [108].

**Figure 24 materials-15-04495-f024:**
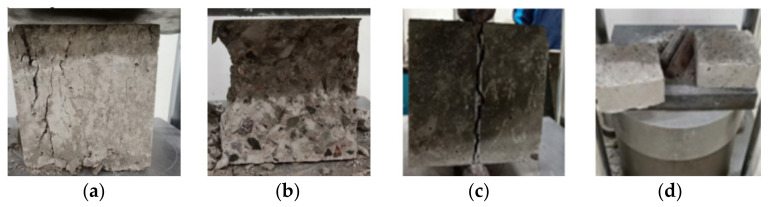
The damaged form of a concrete test block mixed with coal gangue and plant fibers [112]: (**a**) Add plant fiber cube; (**b**) No plant fiber cube compression; (**c**) Add plant fiber cube split; (**d**) No plant fiber cube compression.

**Table 1 materials-15-04495-t001:** Basic physical properties of coal gangue (rock coal gangue) from different locations in China.

Samples	Apparent Density/(kg/m^3^)	Bulk Density/(kg/m^3^)	Water Absorption/%	Crushing Value/%	Void Ratio/%	Ref.
Xianyang City	1960	-	3.98	20.2	-	[18]
Shenmu City	2106	-	8.7	19	-	[19]
Fuxing City	2653	-	3.15	9.9	49.49	[20]
Liuan City	2452	1364	3.3	16	44	[21]
Xuzhou City	2712	-	1.7	16.8	-	[22]
Ningxia City	2760	1560	1.2	-	49.2	[23]
Beijing City	2640	-	1.8	22.6	-	[24]

**Table 2 materials-15-04495-t002:** Chemical composition and mineral composition of different types of coal gangue.

Sources	SiO_2_	Al_2_O_3_	Fe_2_O_3_	MgO	CaO	Na_2_O	K_2_O	TiO_2_	Mineral Composition	Ref.
Clay gangue	59.44	23.43	32.98	2.52	1.06	-	3.47	1.34	Kaolinite, Quartz, Mullite, Calcite	[29]
Sandstone gangue	64.79	18.00	3.82	1.42	4.18	1.62	4.55	-	Kaolinite, Quartz, Illite, Peridotite	[30,31]
Aluminum gangue	42.17	48.41	0.07	0.94	3.77	-	-	1.35	Kaolinite, Quartz, Calcite, Pyrite	[32]
Calcareous gangue	21.71	4.82	4.43	1.61	64.71	0.64	-	0.36	-	[33]

**Table 4 materials-15-04495-t004:** Freeze–thaw damage parameters of gangue concrete fracture toughness and fracture energy under different gangue replacement rates.

Type	Replacement Rate (%)	a	b	R^2^
Fracture toughness (y)	0	0.04016	1.48157	0.99729
	40	0.0402	1.50058	0.99408
	70	0.05165	1.5124	0.99549
	100	0.06214	1.5203	0.99741
Fracture energy (y)	0	0.1859	1.07341	0.99795
	40	0.19664	1.10694	0.99687
	70	0.2146	1.13443	0.99888
	100	0.2457	1.14507	0.9993

## Data Availability

All the relevant data and models used in the study have been provided in the form of figures and tables in the published article.

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
