# Peer review of "Using Chinese Coal Gangue as an Ecological Aggregate and Its Modification: A Review"

_materials, 2022, doi:10.3390/ma15134495_

Round 1

Reviewer 1 Report

The authors did most of suggested corrections in the previous reviewing rounds. But still some typing problems are existing. Please check revised MS in the attachements. I could suggest a minor revision

Author Response

The authors did most of suggested corrections in the previous reviewing rounds. But still some typing problems are existing. Please check revised MS in the attachements. I could suggest a minor revision:

We would like to thank Reviewers for taking the time and effort necessary to review the manuscript. We sincerely appreciate all valuable comments and suggestions, which helped us to improve the quality of the manuscript.

Please check revised MS in the attachements. I could suggest a minor revision.

Thank you very much for your revision suggestions, which make the article more perfect. I have carefully read the modifications you made in the attachment and made improvements to every part of the article according to your marks and notes.

Reviewer 2 Report

Since I reviewed the previous versions of this manuscript, I can see some improvements however there are many outstanding issues:

1. The authors have removed the literature search mechanism from the paper which makes it weak and goes against the standards of review papers. While another reviewer may have suggested this as explained by the authors, I humbly disagree with the other reviewer. A detailed search mechanism, search strings, review mechanism with timeline, method of analysis, and validation make a review paper acceptable. In the absence of these, the work is never reproducible. Therefore the authors must add these details. Please refer to my previous comments and add a detailed literature retrieval method to the paper.

2. The threat to validity explanation is not what was expected. The authors claimed, "In the selection of literatures in this paper, the literatures with high citation rate should be adopted first, and the issues of downloads and reviews should be considered to ensure the validity of the literatures evaluated secondly." This explanation is more about paper selection and not address the threat to validity. I believe the authors misunderstood the term. Threat to validity means ensuring how the eight threats to internal validity: history, maturation, instrumentation, testing, selection bias, regression to the mean, social interaction, and attrition are treated in the published work. Please revise and add this information to the paper. Another way to deal with this is to use systematic review methods such as PRISMA.

3. Line 71 please remove the 1. before (1).

4. Figure 3 starts with the number 2 what is 1? This is confusing. Please add a proper method section and move this figure along with the details requested in point 1 to make it meaningful. I believe this (fig 3) is an overview of the method followed in the paper. It will only make sense if moved to a method section.

Reviewer 3 Report

The paper "Coal gang aggregate ecological concrete and its modification: A review" is interesting and presents a scope that adheres to the proposal of this journal, however modifications are necessary:

(1) The abstract should present more relevant and innovative information. The statements made regarding porosity and durability are obvious and do not add new findings in the literature;

(2) The authors should highlight at the end of the introduction the real innovation of this research, what makes it different from other reviews of this topic?

(3) There are still some works that need to be added, such as: 10.1016/j.cscm.2021.e00751; 10.1016/j.cscm.2022.e01039; 10.3390/app11073036.

(4) Other interactive tables and with results from other researches must be created by the authors, as well as experimental flowcharts;

(5) Conclusion cannot be combined with discussions, separate this into separate topics, furthermore, add a topic of future work and gaps in the literature on the topic;

(6) Try to be more critical in some topics covered, this is important in a review paper.

Author Response

The paper "Coal gang aggregate ecological concrete and its modification: A review" is interesting and presents a scope that adheres to the proposal of this journal, however modifications are necessary:

We would like to thank Reviewers for taking the time and effort necessary to review the manuscript. We sincerely appreciate all valuable comments and suggestions, which helped us to improve the quality of the manuscript.

Point 1: The abstract should present more relevant and innovative information. The statements made regarding porosity and durability are obvious and do not add new findings in the literature;

Response 1: In lines 14 - 17 of the abstract, we first supplement the innovations of this paper. There are few studies on pore characteristics of coal gangue aggregate, so we refer to the existing literature and supplement it in line 352 - 362 of the paper.

Point 2: The authors should highlight at the end of the introduction the real innovation of this research, what makes it different from other reviews of this topic?

Response 2: Thanks very much for your comments. According to the opinions of previous reviewers, the deficiencies of existing literature are pointed out and the innovations of this paper are introduced in line 68-77 as “However, there are few reviews on the coal gangue concrete, especially the topics about preparation methods of coal gangue aggregate, the effect of coal gangue aggre-gate on concrete and the prediction model of coal gangue concrete, as well as the modification materials of gangue aggregate concrete have not been summarized in detail. This paper fully analyzes and summarizes a number of literatures about coal gangue aggregate concrete and achieves the following objectives: (1) to summarize the properties and preparation method of the coal gaugue aggregate, (2) to study the effect of gangue aggregate on concrete performance and prediction model of coal guague concrete, and (3) to review the studies on the influence of the modified materials on coal guague concrete.. In the discussion section, lines 718 to 726 are compared with references.

Point 3: There are still some works that need to be added, such as: 10.1016/j.cscm.2021.e00751; 10.1016/j.cscm.2022.e01039; 10.3390/app11073036.

Response 3: I have carefully read the literature you recommended, and the two references [76][97]were added in this paper, and the works have been added in the revised manuscript as follows:

10.1016/j.cscm.2021.e00751: Lines 351 – 356: With the increase of gangue aggregate, the porosity of low water-binder ratio (0.39) decreases while that of high water-binder ratio (0.50) increases. Low water-binder ratio results in dense interfacial transition zon, which is released into the surrounding interfacial transition zone in the later stage, further promoting the hydration of cement particles. However, in the coagulated soil with high water-binder ratio, coal gangue aggregate has a weak water regulation effect [76].

10.1016/j.cscm.2022.e01039: Lines 507 – 510: As a fine aggregate, FA can effectively improve the gradation performance of the mixture. Short-term (3D /7d) strength increased with the increase of FA content, while long-term (28d) strength and slump increased first and then decreased [97].

10.3390/app11073036: This article uses quartz powder aggregate, does not conform to coal gangue aggregate concrete.

In addition, add two references[75][77] in lines 349 to 359, add the following: 1. Pore structure an essential issue to study the influence of coal gangue content on concrete durability [75]. 2. In addition, proper increase of gangue aggregate can change the pore structure, increase the number of harmful holes and harmful holes, and slightly increase the pore size, reduce the compactness of concrete, and reduce the durability to a certain extent [77].

Point 4: Other interactive tables and with results from other researches must be created by the authors, as well as experimental flowcharts;

Response 4: All charts cited in this paper are based on data from other literatures, and the graphs are created by the authors, and the fitting curves are added according to the comments of reviewers. Graphic references are not directly cited.

Point 5: Conclusion cannot be combined with discussions, separate this into separate topics, furthermore, add a topic of future work and gaps in the literature on the topic;

Response 5: According to the suggestions, this paper divides the discussion, conclusion and prospect into three independent parts, of which chapter 6 is discussion and shows the limitations and gaps of this paper, Chapter 7 is conclusion and Chapter 8 is prospect, which shows the future work.

Point 6: Try to be more critical in some topics covered, this is important in a review paper.

Response 6: The deficiencies of this paper are discussed in lines 727-742 of the literature, and the limitations of this paper are added in lines 785-790 of the conclusions based on the comments of previous reviewers.

[75] Wang, Q.; Huang, C.Y.; Liu, S.; Liu, Y.Q. Research on frost resistance of the coal gangue instead of coarse aggregate concrete. Concrete, 2015(09): 77-79 (In Chinese).

[76] Chen, P.Y.; Zhang, L.H.; Wang, Y.H.; Fang, Y.; Zhang, F.; Xu, Y. Environmentally friendly utilization of coal gangue as aggregates for shotcrete used in the construction of coal mine tunnel. Case Studies in Construction Materials, 2021, 15.

[77] Qiu, J.S.; Zhang, R.Y.; Hou, B.W.; Guan, X.; Gao, X.J.; Li, L.L. Pore Structure Characteristics and Chloride Ion Corrosion Resistance Mechanism of Coal Gangue Concrete under Dry-Wetting Cycles. Bulletin of the Chinese Ceramic Society, 2021, 40(12): 3993-4001 (In Chinese).

[97] Yin, S.H.; Yan, Z.P.; Chen, X.; Wang, L.M. Effect of fly-ash as fine aggregate on the workability and mechanical properties of cemented paste backfill. Case Studies in Construction Materials, 2022, 16.

Round 2

Reviewer 2 Report

Thank you for addressing my comments.

Reviewer 3 Report

ok

This manuscript is a resubmission of an earlier submission. The following is a list of the peer review reports and author responses from that submission.

Round 1

Reviewer 1 Report

Thank you for providing me with the opportunity to read “Coal gangue aggregate ecological concrete and its modification: A review”. I have the following comments:

  • Please check the paper formatting. Some sections (conclusion etc.) have different formatting.
  • The abstract doesn’t show what databases are used, what timeline is followed, and how the review is conducted. Please add such important details. In addition, no systematic search methodology is mentioned in the abstract; this information is important for the type of article produced.
  • Please provide references for the percentages and numbers used in the first paragraph of the introduction (7.742 billion tons, 1.2%, 50%, 75%).
  • The introduction has an organization problem. The authors seem confused about what message to convey with the introduction, which makes the text lack cohesion and coherence. To better structure the introduction, I suggest that the authors rewrite the following basic structure: context, motivation and justification, problem, methodology used in the solution, evaluation, main results, conclusion, and article structure.
  • The novelty of the paper is not clear from the introduction. What are the key innovations here? What is new, and what are the additions to the body of knowledge made through this study? This is a serious weakness of the paper. Please revise carefully.
  • Please add the objectives of the study in a numbered format to the introduction and later in the discussion, refer back to them and discuss how these are achieved.
  • For comparing the physical properties of coal gangue, a case of China is not enough. The authors need to provide examples of other countries as well and compare them with China in Table 1.
  • Figure 4 needs more explanation. What are the key findings in this figure? It has a single line mention which needs more explanation. Alternatively, if it is not adding much value, please remove it.
  • The paper claims to be a review paper; however, there is no mention of how the review was conducted. No systematic review and guidelines were applied to the selection, analysis, and report process, which can bring problems to the reproducibility of the process and reliability of the collected and selected information. What databases are used, and what is the search string used in databases to retrieve relevant papers? Please have a look at some recent review papers and revise the article properly:
    • https://www.sciencedirect.com/science/article/abs/pii/S0950584915000646
    • https://www.mdpi.com/2071-1050/10/9/3142
  • What were the inclusion and exclusion criteria for articles in the selection phases? How were the selected studies evaluated? Were there any duplicate articles? If yes, how were these addressed?
  • In the absence of a holistic review mechanism understanding the article and how it contributes new knowledge to the body of knowledge is not clear at all. I suggest the authors revise the article thoroughly and justify their review method before presenting any results.
  • The discussion section must be moved ahead of the conclusion and should be extended. Further, it must be compared with other similar studies to highlight its novelties and key innovations. In the absence of this important section, the paper remains very weak. Please revise.
  • Please clearly add the limitations of the study to the conclusion section.
  • The future directions for expanding upon this study are not clear. Please revise

Author Response

Response to Reviewer 1 Comments

Thank you for providing me with the opportunity to read “Coal gangue aggregate ecological concrete and its modification: A review”. I have the following comments.

We would like to thank Reviewers for taking the time and effort necessary to review the manuscript. We sincerely appreciate all valuable comments and suggestions, which helped us to improve the quality of the manuscript.

Point 1: Please check the paper formatting. Some sections (conclusion etc.) have different formatting.

Response 1: The format of the paper has been modified and checked correctly.

Point 2: The abstract doesn’t show what databases are used, what timeline is followed, and how the review is conducted. Please add such important details. In addition, no systematic search methodology is mentioned in the abstract;  this information is important for the type of article produced.

Response 2: Thanks very much for your advices! This paper was searched in Scopus, Elsever and China National Knowledge Internet database in April 2022. Filter by search presence of terms, considered period, publication form of paper, and the searching area. This information has been added in line 13-15 and shown as red in the revised manuscript.

Point 3: Please provide references for the percentages and numbers used in the first paragraph of the introduction (7.742 billion tons, 1.2%, 50%, 75%).

Response 3: The references [1] and [2] have been added in Line 34-43: "7.742 billion tons" and "1.2%" from the reference[1]; "50%" and "75%" from the reference[2].

Point 4: The introduction has an organization problem. The authors seem confused about what message to convey with the introduction, which makes the text lack cohesion and coherence.  To better structure the introduction, I suggest that the authors rewrite the following basic structure: context, motivation and justification, problem, methodology used in the solution, evaluation, main results, conclusion, and article structure.

Response 4: Thank you very much for your suggestions! In order to express the information more accurately, the topic is changed to "Using Chinese coal gangue as an ecological aggregate and their modification: A review ".

The overall framework of the introduction is reframed according to the following orgnization. The introduction begins with the first data collection of coal consumption in China and the world, which leads to the coal gangue pollution situation of China,it is necessary to deal with coal gangue and make it widely used in the construction industry. Second and upon proposed the insufficient research on coal gangue concrete, especially the topic about preparation methods of coal gangue aggregate, the effect of gangue aggregate on concrete and the prediction model of coal gangue concrete, as well as the modification materials of gangue aggregate concrete,which are the innovations of this paper. Third, research objectives were proposed to study the properties and preparation method of the coal guague aggregate, present the effect of gangue aggregate on concrete performance and prediction model and study the influence the modi-fied materials on coal gangue concrete. To address this, the systematic search methodology was employed. All the literatures were searched in Scopus, Elsever and China National Knowledge Internet database before April 2022 andfiltered by search presence of terms, considered period, publication form of paper, and the searching area. Lastly,the paper structure was given.

Article structure: This paper begins with an introduction of the systematic search methodology. Second and upon concluded the physical and chemical properties of gangue aggregate, the coal gangue aggregate preparation methods were summarized. Third, studied the per-formance of coal gangue aggregate to concrete, including working performance, me-chanical properties, durability and carbonation resistance. Then, the influence of modi-fied materials on coal gangue concrete performance was studied. Lastly,discussion and conclusion was given to provide effective theoretical support to the research of coal gangue concrete.

Point 5: The novelty of the paper is not clear from the introduction. What are the key innovations here? What is new, and what are the additions to the body of knowledge made through this study?  This is a serious weakness of the paper. Please revise carefully.

Response 5: The deficiencies of the existing papers was pointed out, which are the innovations of this paper and the key research contents different from other studies. “However, there are few reviews on the coal gangue concrete, especially the topics about preparation methods of gangue aggregate, the effect of gangue aggregate on concrete and the modification materials of gangue aggregate concrete have not been summarized in detail.” has been added in line 84-87.

Point 6: Please add the objectives of the study in a numbered format to the introduction and later in the discussion, refer back to them and discuss how these are achieved.

Response 6: Thank you very much for your suggestions! The objectives of this review are added in line 87~90 as follows: 1. Study the properties and preparation method of the coal gaugue aggregate. 2. Study the effect of coal gangue aggregate on concrete performance and prediction model of coal guague concrete . 3. Study the influence of the modified materials on coal guague concrete.

The conclusions has been revised in order to refer back to the above abjectices in line 87-90.

Point 7: For comparing the physical properties of coal gangue, a case of China is not enough. The authors need to provide examples of other countries as well and compare them with China in Table 1.

Response 7: Most of the literature on coal gangue aggregate concrete is from China, so the title is changed to “Using Chinese coal gangue as an ecological aggregate and their modification: A review” according to the overall content.

Point 8: Figure 4 needs more explanation. What are the key findings in this figure? It has a single line mention which needs more explanation. Alternatively, if it is not adding much value, please remove it.

Response 8: Thanks very much for your suggestions! Two explanations of Figure 4 has been added Line 163-166 in the manuscript. First, coal gangue has a high content of silica and aluminum, but most of them are crystalline phase and have no activity. Second, Calcination can effectively generate metakaolinite active substance. This article has carried on the supplement.

Point 9: The paper claims to be a review paper; however, there is no mention of how the review was conducted. No systematic review and guidelines were applied to the selection, analysis, and report process, which can bring problems to the reproducibility of the process and reliability of the collected and selected information. What databases are used, and what is the search string used in databases to retrieve relevant papers? Please have a look at some recent review papers and revise the article properly:

https://www.sciencedirect.com/science/article/abs/pii/S0950584915000646

https://www.mdpi.com/2071-1050/10/9/3142

Response 9: Thanks very much for your suggestions, the refrences you provided is very voluable for the revision of the manuscript. The systemtic review methods were added Line 107-120: in the Chapter 2. Thanks again for your help.

Point 10: What were the inclusion and exclusion criteria for articles in the selection phases? How were the selected studies evaluated? Were there any duplicate articles? If yes, how were these addressed?

Response 10: Inclusion criteria: 1. Clearly defined articles are found according to the search characters and review process, which are in line 106-120 with the part we are studying. 2. The paper is reasonably structured and logically rigorous, with sufficient experiments and real and effective data. 3. The material properties of each part are similar to the research methods. Exclusion criteria: 1. Duplicate literature. 2. There are design defects and poor quality. 3. Incomplete experimental data, unreasonable discussion and analysis.

Evaluation of the selected research: 1. The research has certain value to the development of the discipline and has sufficient theoretical basis. 2. Most of the literatures were published in recent 10 years, the research status can be known well. 3. Clear objective, appropriate content, and certain innovative ideas; 4. Clear expression of academic concepts.

There are no duplicates.

Point 11: In the absence of a holistic review mechanism understanding the article and how it contributes new knowledge to the body of knowledge is not clear at all. I suggest the authors revise the article thoroughly and justify their review method before presenting any results.

Response 11: This article has been revised significantly. The methods of review are given in lines 118-120 in Chapter 2. The structure of the article is reorganized, and the structure and content of the article have been modified according to the review method from the line 112-125, which proves the review method. Moreover, the innovative of this review has been given in line 84-87 in the introduction

Point 12: The discussion section must be moved ahead of the conclusion and should be extended. Further, it must be compared with other similar studies to highlight its novelties and key innovations. In the absence of this important section, the paper remains very weak. Please revise.

Response 12: The discussion section has moved ahead of the conclusion section in line 840-847: The literature [3,4] has similar conclusions, but the study lacks the preparation method of coal gangue, the effect of gangue aggregate on concrete performance, the performance prediction model of concrete and the summative description of modified materials. In this paper, the preparation method of concrete is added to further explain the influence of gangue aggregate on concrete performance, the effect of gangue aggregate on concrete is studied fully, and the performance prediction model is proposed, which is beneficial to engineering application. In addition, the influence of modified materials on gangue aggregate concrete performance is summarized and described, providing ideas for scholars' research.

Point 13: Please clearly add the limitations of the study to the conclusion section.

Response 13: The limitations of this study lie in the precipitation of heavy metal ions in gangue aggregate concrete, the error of constitutive model, the use of nanomaterials, the lack of high-grade concrete and its combination with engineering practice. These limitations have been added in line 851-863 in conclusion section

Point 14: The future directions for expanding upon this study are not clear. Please revise

Response 14: The future research direction has been revised in the line 962-975 in section 7.3. Due to the chemical characteristics of gangue aggregate and its influence on concrete degree surface, heavy metal leaching analysis, constitutive model establishment, utilization of nanomaterials, and exploration of high performance concrete combined with engineering practice are the directions of future development.

[1] Yang, Y. Energy globalization of China:Trade, investment, and embedded energy flows. Journal of Geographical Sciences, 2022, 32(03): 377-400.

[2] Li, J.Y.; Wang, J.M. Comprehensive utilization and environmental risks of coal gangue: A review. Journal of Cleaner Production, 2019, 239, 117946.

[3] Wang, A.G.; Zhu, Y. Y.; Xu, H.Y.; Liu, K.W.; Jing, Y.; Sun, D.S. Research Progress on Coal Gangue Aggregate for Concrete. Bulletin of the Chinese Ceramic Society, 2019, 38(07): 2076-2086. (In Chinese)

[4] Gao, S.; Zhang, S.M.; Guo, L.H. Application of Coal Gangue as a Coarse Aggregate in Green Concrete Production: A Review. Materials, 2021, 14(22), 6803.

Attached is the revised manuscript.

Reviewer 2 Report

The MS, titled "Coal gangue aggregate ecological concrete and its modification: A review", reports a compressive review of Chinese coal gangue as an alternative aggregate. Since China will be the biggest coal mining country in the following three decades, using such materials will have an important impact on reducing the environmental impact of coal mining and the cement industry. The report data in the MS almost summarizes the latest studies from China. Nevertheless, the authors try to avoid using dense "Hao et al. (2023) done this" or "Guo (2000) studied this" phrases. Since the MS is a review paper, the authors try to provide more literature review, as they did in section 6.3. Furthermore, the MS is mainly based on a review of Chinese coal gangue as an ecological concrete. Therefore, I could suggest the authors revise the title as "Using Chinese coal gangue as an ecological aggregate and their modification: A review" or try to enlarge their study by including other coal gangues in other countries. Finally, the authors should try to shorten the MS as much as they can. I noted several typos or writing style suggestions in the attached revised MS. Overall, I could suggest a moderate revision (which is not a choice in the review system), and would like to re-consider MS after suggested corrections are made.

 Since China will be the biggest coal mining country in the following three decades, using such materials will have an important impact on reducing the environmental impact of coal mining and the cement industry. The report data in the MS almost summarizes the latest studies from China. Nevertheless, the authors try to avoid using dense "Hao et al. (2023) done this" or "Guo (2000) studied this" phrases. Since the MS is a review paper, the authors try to provide more literature review, as they did in section 6.3. Furthermore, the MS is mainly based on a review of Chinese coal gangue as an ecological concrete. Therefore, I could suggest the authors revise the title as "Using Chinese coal gangue as an ecological aggregate and their modification: A review" or try to enlarge their study by including other coal gangues in other countries. Finally, the authors should try to shorten the MS as much as they can. I noted several typos or writing style suggestions in the attached revised MS. Overall, I could suggest a moderate revision (which is not a choice in the review system), and would like to re-consider MS after suggested corrections are made.

Author Response

Response to Reviewer 2 Comments

The MS, titled "Coal gangue aggregate ecological concrete and its modification: A review", reports a compressive review of Chinese coal gangue as an alternative aggregate. Since China will be the biggest coal mining country in the following three decades, using such materials will have an important impact on reducing the environmental impact of coal mining and the cement industry. The report data in the MS almost summarizes the latest studies from China. 

 Since China will be the biggest coal mining country in the following three decades, using such materials will have an important impact on reducing the environmental impact of coal mining and the cement industry. The report data in the MS almost summarizes the latest studies from China. 

We would like to thank Reviewers for taking the time and effort necessary to review the manuscript. We sincerely appreciate all valuable comments and suggestions, which helped us to improve the quality of the manuscript.

Point : Nevertheless, the authors try to avoid using dense "Hao et al. (2023) done this" or "Guo (2000) studied this" phrases. Since the MS is a review paper, the authors try to provide more literature review, as they did in section 6.3. Furthermore, the MS is mainly based on a review of Chinese coal gangue as an ecological concrete. Therefore, I could suggest the authors revise the title as "Using Chinese coal gangue as an ecological aggregate and their modification: A review" or try to enlarge their study by including other coal gangues in other countries. Finally, the authors should try to shorten the MS as much as they can. I noted several typos or writing style suggestions in the attached revised MS. Overall, I could suggest a moderate revision (which is not a choice in the review system), and would like to re-consider MS after suggested corrections are made.

Response : According to the comments of reviewers, I changed the topic to "Using Chinese coal gangue as an ecological aggregate and their modification: A review". At the same time, thank you very much for improving the errors and style of the content of the article, which has been modified according to the requirements of the reviewer.

MS has been modified and supplemented according to the requirements of the attachment. The required reply is as follows:  

Point : Geological age does not have any influnce on mineralogical or elemental compositions of coal gangues!

Response :There are few studies on the influence of geological time on coal gangue, but some scholars have studied it, For example, literature [1] has studied relevant topic.

In addition, all the pictures in the article that are not marked with literature sources are made and photographed by ourselves, not from literature.

[1] Gu, B.W.; Wang, P.M. Analysis of Factors Affecting Pozzolanic Activity in Thermal Activated Coal Gangue. Journal of Building Materials, 2009, 12(01), 6-11.

Attached is the revised manuscript.

Reviewer 3 Report

The paper is about reviewing published work on coal gangues, properties and applications. The authors presented an interesting work, but it requires numerous improvements to be of satisfactory quality.

General remarks

Make clear the purpose of the review in the introduction.

Lack of numbering of lines, hence it is impossible to precisely indicate the reservations.

Combine chapters 4 and 5

Do not submit the literature list as a superscript, but as normal text.

The work is a review article, it draws attention to the lack of world literature, it is only Chinese. Add or replace literature.

Detailed comments

  1. To the carbon consumption data and Fig. 1, add the appropriate literature.
  2. Coal gangue is a kind of hard rock produced during coal formation - elaborate on. It is essential for work. The hard grains in the coal may have been formed during the carbonization process, the coal gangue may take the form of carbon overgrowths and also get into the coal from the rocks accompanying the coal.
  3. Coal gangue is the waste of coal combustion - this sentence partially contradicts the above, you need to be clear about what you mean by coal gangues - non-combustible parts of coal, rock waste or products of combustion. There is a definition: "Coal gangue is a commercially worthless material that closely attaches to or surrounds coal in coalfield."
  4. Point 2.3 Waste rock can also be formed by claystones, they have a layered structure. Include it in the subsection.
  5. Spontaneous combustion coal gangue - waste rock changes color after being stored in heaps, during which the coal parts are burnt. You need to make it clear at work.
  6. Fig. 11, 12, 13, 14, 15, 16, 17, 19, 20, 21, 22 - do not connect measurement points, just add regression lines
  7. Add the Conclusion chapter at the end of the work
  8. The contents of chapter 9 Discussion and Outlook should be added to chapter 7 and the title itself removed.

Author Response

Response to Reviewer 3 Comments

The paper is about reviewing published work on coal gangues, properties and applications. The authors presented an interesting work, but it requires numerous improvements to be of satisfactory quality.

We would like to thank Reviewers for taking the time and effort necessary to review the manuscript. We sincerely appreciate all valuable comments and suggestions, which helped us to improve the quality of the manuscript.

General remarks:

Point 1: Make clear the purpose of the review in the introduction.

Response 1: Line 89-92: Thank you very much for your suggestion! The review purpose are concluded as follows, and these have been modified accordingly in the introduction.

By summarizing and evaluating a large number of literatures, the objectives are as follows: 1. Explore the preparation method of the latest aggregate. 2. Study the effect of gangue aggregate on concrete. 3. Study on improving performance of coal gangue concrete with modified materials.

Point 2: Lack of numbering of lines, hence it is impossible to precisely indicate the reservations.

Combine chapters 4 and 5

Response 2: The line number has been added.Changes have been made to merge.

Point 3: Do not submit the literature list as a superscript, but as normal text.

Response 3: Modifications have been made and the superscript changed to normal text.

Point 4: The work is a review article, it draws attention to the lack of world literature, it is only Chinese. Add or replace literature.

Response 4: Line 1: Most of the researches on coal gangue aggregate concrete are conducted by Chinese scholars. Therefore, according to the opinions of the second reviewer, The title of the article was changed to "Using Chinese coal gangue as an Ecological Aggregate and their modification: A review".

Detailed comments:

Point 1: To the carbon consumption data and Fig. 1, add the appropriate literature.

Response 1: Line 34-43: Sources and literatures have been added to the data of carbon consumption. The data in Figure 1 is from the International Energy Agency. url https://www.iea.org/countries. Data sources have been added.

Point 2: Coal gangue is a kind of hard rock produced during coal formation - elaborate on. It is essential for work. The hard grains in the coal may have been formed during the carbonization process, the coal gangue may take the form of carbon overgrowths and also get into the coal from the rocks accompanying the coal.

Response 2: Line 47-51: Thank you very much for your comments. In the introduction part of the article, I have added the following sentence for a better explanation: “Coal gangue is a kind of hard rock produced during coal formation, which is formed at the same timewith coal seam. The hard grains in the coal may have been formed during the carbonization process, the coal gangue may take the form of carbon overgrowths and also get into the coal from the rocks accompanying the coal.   However, most of it is limestone. Due to the long-term penetration and diffusion of limestone by coal seams, it contains hard grains of coal in black and gray.”

Point 3: Coal gangue is the waste of coal combustion - this sentence partially contradicts the above, you need to be clear about what you mean by coal gangues - non-combustible parts of coal, rock waste or products of combustion. There is a definition: "Coal gangue is a commercially worthless material that closely attaches to or surrounds coal in coalfield."

Response 3: Coal gangue is the waste of coal combustion -- I have deleted this sentence, which is wrong about coal gangue, and the definition of coal gangue is supplemented and revised in Point 2.

Point 4: Point 2.3 Waste rock can also be formed by claystones, they have a layered structure.  Include it in the subsection.

Response 4: Line 156-161: Thank you very much for your comments! the inroduction about claystones has been added in the Line 156-161 in the manuscript .The article has been supplemented as follows: According to the content of oxide in coal gangue, coal gangue can be divided into clay gangue (SiO2(40%-70%), Al2O3(15%-30%)), sandstone gangue (SiO2 > 70%), aluminum gangue (Al2O3 > 40%) and calcareous gangue (CaO > 30%). Clay coal gangue is one of the most commonly used coal gangue aggregate concrete. It has high silicon aluminum material, layered structure and can produce activity under certain conditions. Table 2 shows the chemical composition of coal gangue in different areas. The content of SiO2 and Al2O3, and in most coal gangue is more than 70%, which is considered to have a certain volcanic ash effect.

Point 5: Spontaneous combustion coal gangue - waste rock changes color after being stored in heaps, during which the coal parts are burnt. You need to make it clear at work.

Response 5: Line 232-234: In the modified section 4.1.2 of spontaneous combustion of coal gangue, the supplementary explanation is as follows: The color of spontaneous combustion coal gangue will change after combustion. According to the content of iron oxide, the color will become gray white, yellow white or red.

Point 6: Fig. 11, 12, 13, 14, 15, 16, 17, 19, 20, 21, 22 - do not connect measurement points, just add regression lines.

Response 6: All modifications have been made as required.

Point 7: Add the Conclusion chapter at the end of the work.

Response 7: Line 870-967: A supplement has been added at the end of the article.

Point 8: The contents of chapter 9 Discussion and Outlook should be added to chapter 7 and the title itself removed.

Response 8: Line 840-981: I wonder if you mean the contents of chaper 9 shoud be added to chapter 8? Because the seventh chapter of this paper studies the influence of coal gangue aggregate concrete from the perspective of different admixtures, which is the research part of the paper. So I have removed chaper 9 Disscussion and Outlook, and added to chaper 8. Please check it, thank you very much.  

Round 2

Reviewer 1 Report

Thank you for addressing some of my comments. The following still needs your attention.

1. While the authors have added the literature search mechanism as previously suggested, the exact search strings used in the paper are missing. Please refer to my last comment and add the exact search strings in line with the suggested references. A table should be added for this with numbers against each search string for the mentioned search engines. These must be added to ensure that the study is reproducible. 

2. It is not clear how many articles were reviewed what was the distribution of various article types? 

3. At the end of section 1, please add a paragraph about how the study is organized as it is hard to follow.

4. How were the threats to validity tackled? There is no mention of this in the paper. It is important to ensure a systematic approach is followed in the retrieval and review of the articles.

5. Figure 3 is not readable, please add a readable version.

Author Response

We would like to thank Reviewers for taking the time and effort necessary to review the manuscript. We sincerely appreciate all valuable comments and suggestions, which helped us to improve the quality of the manuscript.

Reviewer 2 Report

The authors did some of suggested corrections in the first round. The MS is very long and not see to follow. I highly recommend to the authors summarize several parts in the section 4. Furthermore, the authors jumped one point to another point, then turn back again. The MS sounds like just combination of several research paper and the recent format is not looking a review. The authors could summarize in a brief way firstly what is coal gangue and types, then physical, chemical and mineralogical compositions of each mentioned coal gangues in China, and provide a breif review of applicable of coal gangue. Please check my notes in the first round and attached copy in this round. I could advise to the author re-consider this MS and re-organize as a review article, and re-submit. 

Author Response

Response to Reviewer 2 Comments

The authors did some of suggested corrections in the first round.  The MS is very long and not see to follow.  I highly recommend to the authors summarize several parts in the section 4.  Furthermore, the authors jumped one point to another point, then turn back again.  The MS sounds like just combination of several research paper and the recent format is not looking a review.  The authors could summarize in a brief way firstly what is coal gangue and types, then physical, chemical and mineralogical compositions of each mentioned coal gangues in China, and provide a breif review of applicable of coal gangue.  Please check my notes in the first round and attached copy in this round.  I could advise to the author re-consider this MS and re-organize as a review article, and re-submit.

We would like to thank Reviewers for taking the time and effort necessary to review the manuscript. We sincerely appreciate all valuable comments and suggestions, which helped us to improve the quality of the manuscript.

The authors did some of suggested corrections in the first round.  The MS is very long and not see to follow.  I highly recommend to the authors summarize several parts in the section 4. Furthermore, the authors jumped one point to another point, then turn back again.

Response: Thank you very much for your suggesstions! several parts have been summerized in the section 4(which has been revised to section 3 in the revised manuscript): Therefore, combined with the above mentioned, undisturbed coal gangue can be directly used as aggregate to prepare concrete. After combustion, the physical properties of spontaneous combustion coal gangue are reduced to a certain extent, but the direct crushing of coal gangue into aggregate still meets the requirements of preparing C30 concrete. Coal gangue has a large number of active components, high crystallinity, does not have the activity, through mechanical activation, calcination activation (700℃-750℃), chemical activation and composite activation can effectively improve the activity of coal gangue; High temperature calcination (930℃) of coal gangue can produce ceramsite,, The micro-pump effect of ceramsite can effectively improve the gangue concrete.  

In order to make the section 4 (section 3 in the revised munuscript) more clear, “4.3 preparation of ceramsite from coal gangue” has been changed to”3.3 high temperature calcination of coal gangue”, which, together with direct utilization and activation of coal gangue, these are three preparation methods of coal gangue aggregate.   

The MS sounds like just combination of several research paper and the recent format is not looking a review.

Response: Significant revisions have been made to the full text, as shown in the attachment. All the phrases "Hao et al. (2023) done this" or "Guo (2000) studied this" have been revised, some contents repeated and insignificant cotents have been deleted, and the summary of each part have been given. Please check it, Thank you.

The authors could summarize in a brief way firstly what is coal gangue and types, then physical, chemical and mineralogical compositions of each mentioned coal gangues in China, and provide a breif review of applicable of coal gangue.

Response: Thank you very much for your suggestions! Coal gangue is defined in line150-152. Because of the coal gangue is classified based on the chemical composition, so the types of coal gangue has been added in lines 150 to 155 in section 2.2, and the for the table is added as follows:

According to the content of oxide in coal gangue, coal gangue can be divided into clay gangue (SiO2(40%-70%), Al2O3(15%-30%)), sandstone gangue (SiO2 > 60%), aluminum gangue (Al2O3 > 40%) and calcareous gangue (CaO > 30%) [12]. Table 2shows the chemical composition and mineral composition of different types of coal gangue. Clay gangue is one of the most commonly used coal gangue aggregate concrete. It has high silicon aluminum material, layered structure and can produce activity under certain conditions.

Table 2 Chemical composition and mineral composition of different types of coal gangue.

Sources

SiO2

Al2O3

Fe2O3

MgO

CaO

Na2O

K2O

TiO2

Mineral composition

Ref.

Clay gangue

59.44

23.43

32.98

2.52

1.06

-

3.47

1.34

Kaolinite, Quartz, Mullite, Calcite

[13]

Sandstone gangue

64.79

18.00

3.82

1.42

4.18

1.62

4.55

-

Kaolinite, Quartz, Illite, Peridotite

[14,15]

Aluminum gangue

42.17

48.41

0.07

0.94

3.77

-

-

1.35

Kaolinite, Quartz, Calcite, Pyrite

[16]

Calcareous gangue

21.71

4.82

4.43

1.61

64.71

0.64

-

0.36

-

[17]

Please check my notes in the first round and attached copy in this round.  I could advise to the author re-consider this MS and re-organize as a review article, and re-submit.

Response: 1. All the phrases "Hao et al. (2023) done this" or "Guo (2000) studied this" have been revised. 2. According to the first round of response, the chemical composition of coal gangue was supplemented and the source of coal gangue was supplemented. 3. The full text has been revised and deleted according to the modification suggestions in the reviewer's attachment. The modification result is shown in the attachment.

In order to keep the article smooth, I have modified the full text as follows:

  • lines 168 - 170 have been deleted.
  • lines 217 - 225 have been deleted.
  • This article is modified at lines 280- 282. The following” High temperature calcination is a kind of pottery process, ceramsite produced by high temperature calcination of coal gangue can be adopted as aggregate to prepare concrete.”
  • lines 293 - 294 have been deleted.
  • lines 299 - 300 have been deleted.
  • Make a position shift on lines 300 – 302.
  • lines 325 - 326 have been deleted.
  • lines 334 - 336 have been deleted.
  • lines 439 - 441 have been deleted.
  • lines 486 - 489 have been deleted.
  • lines 540 - 543 have been deleted.
  • Add a summary to lines 553 - 561.
  • lines 617 - 618 have been deleted.
  • lines 625 - 632 have been deleted.
  • lines 643 - 644 have been deleted.
  • lines 652 - 653 have been deleted.
  • lines 682 - 687 have been deleted, and modified.
  • lines 691 - 693 have been deleted.
  • lines 759 - 760 have been deleted.
  • lines 772 - 775 have been deleted, and modified.
  • Add a summary to lines 782- 800.

          Specific modifications are marked in the attachment.

In addition, some modifications are explained below:

According to the suggestions of reviewer 1. Point 3: At the end of section 1, please add a paragraph about how the study is organized as it is hard to follow. Lines 90 to 96 below were not deleted as you requested. Others have been modified according to your suggestions.
